# Limits on Testing Structural Changes in Ising Models

**Aditya Gangrade**
Boston University
gangrade@bu.edu

**Bobak Nazer**
Boston University
bobak@bu.edu

**Venkatesh Saligrama**
Boston University
srv@bu.edu

## Abstract

We present novel information-theoretic limits on detecting sparse changes in Ising models, a problem that arises in many applications where network changes can occur due to some external stimuli. We show that the sample complexity for detecting sparse changes, in a minimax sense, is no better than learning the entire model even in settings with local sparsity. This is a surprising fact in light of prior work rooted in sparse recovery methods, which suggest that sample complexity in this context scales only with the number of network changes. To shed light on when change detection is easier than structured learning, we consider testing of edge deletion in forest-structured graphs, and high-temperature ferromagnets as case studies. We show for these that testing of small changes is similarly hard, but testing of *large* changes is well-separated from structure learning. These results imply that testing of graphical models may not be amenable to concepts such as restricted strong convexity leveraged for sparsity pattern recovery, and algorithm development instead should be directed towards detection of large changes.

## 1 Introduction

Recent technological advances have lead to the emergence of high-dimensional datasets in a wide range of scientific disciplines [YY17; Cos+10; PF95; Bre15; Lok+18; WSD19; Ban18], where the observations are modeled as arising from a probabilistic graphical model (GM), and the goal is to recover the network [Orl+15]. While full network recovery is sometimes useful, and there has been a flurry of activity [DM17; SW12] in this context, we are often interested in *changes* in network structure in response to external stimuli, such as changes in protein-protein interactions across different disease states [IK12] or changes in neuronal connectivity as a subject learns a task [Moh+16].

A baseline approach is to estimate the network at each stage, and then compare the differences. However, such observations exhibit significant variability, and the amount of data available may be too small for this approach to yield meaningful results. On the other hand, *reliably recovering network changes should be easier than full reconstruction*. While prior works have proposed inference algorithms to explore this possibility [ZCL14; XCC15; FB16; BVB16; BZN18; Zha+19; Cai+19], we do not have a good mathematical understanding of when this is indeed easier.

To shed light on this question, we propose to derive information-theoretic limits for two structural inference problems over degree-bounded Ising models. The first is goodness-of-fit testing ($\mathbb{GOF}$). Let $G(P)$ be the network structure (see §2) of an Ising model $P$. $\mathbb{GOF}$ is posed as follows.

> $\mathbb{GOF}$ : *Given an Ising model $P$ and i.i.d. samples from another Ising model $Q$, determine if $P = Q$ or if $G(P)$ and $G(Q)$ differ in at least $s$ edges.*

The second is a related estimation problem, termed error-of-fit ($\mathbb{EOF}$), that demands localising differences in $G(P)$ and $G(Q)$ (if distinct).

> $\mathbb{EOF}$: *Given an Ising model $P$ and i.i.d. samples from another Ising model $Q$ that is either equal to $P$, or has a network structure that differs from that of $P$ in $s$ edges or more, determine the edges where $G(P)$ and $G(Q)$ differ.*

Notice that the above problems are restricted to models that are either identical, or significantly different. 'Tolerant' versions (separating small changes from large) are not pursued here. The main question of interest is: *For what classes of Ising models is the sample complexity of the above inference problems significantly smaller than that of recovering the underlying graph directly?*

**Contribution.** We prove the following surprising fact: up to relatively large values of $s$, the sample complexities of $\mathbb{GOF}$ and $\mathbb{EOF}$ are *not* appreciably separated from that of structure learning ($\mathbb{SL}$). Our bound is surprising in light of the fact that prior works [Liu+14; Liu+17; FB16; KLK19; Cai+19] propose algorithms for $\mathbb{GOF}$ and $\mathbb{EOF}$, and claim recovery of *sparse* changes is possible with sample complexity much smaller than $\mathbb{SL}$. Concretely, for models with $p$ nodes, degrees bounded by $d$, and non-zero edge weights satisfying $\alpha \leq |\theta_{ij}| \leq \beta$ (see §2), the sample complexity of $\mathbb{SL}$ is bounded as $O(e^{2\beta d}\alpha^{-2}\log p)$. We show that if $s \ll \sqrt{p}$, then the sample complexity of $\mathbb{GOF}$ is at least $e^{2\beta d - O(\log(d))}\alpha^{-2}\log p$, and that if $s \ll p$, then the sample complexity of $\mathbb{EOF}$ has the same lower bound. We further show that the same effect occurs in the restricted setting of detecting edge deletions in forest-structured Ising models, and, to some extent, in detecting edge deletions in high-temperature ferromagnets. In the case of forests, we tightly characterise this behaviour of $\mathbb{GOF}$, showing that for $s \ll \sqrt{p}$, $\mathbb{GOF}$ has sample complexity comparable to $\mathbb{SL}$ of forests, while for $s \gg \sqrt{p}$, it is vanishingly small relative to $\mathbb{SL}$. For high-temperature ferromagnets, we show that detecting changes is easier than $\mathbb{SL}$ if $s \gg \sqrt{pd}$, while this does not occur if $s \ll \sqrt{pd}$. These are the first structural testing results for edge edits in natural classes of Ising models that show a clear separation from $\mathbb{SL}$ in sample complexity.

*Technical Novelty.* The lower bounds are shown by constructing explicit and flexible obstructions, utilising Le Cam's method and $\chi^2$-based Fano bounds. The combinatorial challenges arising in directly showing obstructions on large graphs are avoided by constructing obstructions with well-controlled $\chi^2$-divergence on small graphs, and then *lifting* these to $p$ nodes via tensorisation in a process that efficiently deals with combinatorial terms. The main challenge is obtaining precise control on the $\chi^2$-divergence between graphs based on cliques, which is attained by an elementary but careful analysis that exploits the symmetries inherent in Ising models on cliques. The most striking instance of this is the 'Emmentaler clique' (Fig. 2), which is constructed by removing $\Theta(d^2)$ edges from a $d$-clique in a structured way. Despite this large edit, we show that it is exponentially hard (in low temperatures) to distinguish this clique with large holes from a full clique.

## 1.1 Related Work

**Statistical Divergence Based Testing.** Related to our problem, but different from our setup, $\mathbb{GOF}$ of Ising models has been studied under various statistical metrics such as the symmetrised KL divergence [DDK19] and total variation [Bez+19]. More refined results and extensions have appeared in [GLP18; DDK17; Can+17; Ach+18]. These are tests that certify whether or not a particular statistical distance between two distribution is larger than some threshold. In contrast, our focus is on *structural* testing and estimation, namely, whether or not the change in the network is a result of edge-deletions or edge-additions. As such, statistically-based $\mathbb{GOF}$ tests do not have a direct bearing on structural testing. Divergences can be large in structurally irrelevant ways, e.g., if a few isolated nodes in a large graph become strongly interacting, a large KL divergence is induced, but this is not a significant change in the network on the whole (Also see §E.1). In light of applications which demand structure testing as a means to interpret phenomena, and this misalignment of goals, testing in the parameter space is compelling, and testing the network is the simplest instance of this.

**Sparse-Recovery-Based Structural Testing Methods.** More directly related to our work, are those that are based on direct change estimation (*DCE*) [FB16; Liu+14; Liu+17; LFS17; KLK19], which attempt to directly characterize the difference of parameters $\delta^* = \theta_P - \theta_Q$ by leveraging sparsity of $\delta^*$. These works leverage the 'KL Importance Estimation Procedure' (KLIEP), the key insight of which is that the log-likelihood ratios can be written in a form that is suggestive of expressions from sparse-pattern recovery methods, to define the empirical loss function

$$\mathcal{L}(\delta) = -\langle \delta, \hat{\mathbb{E}}_Q[XX^T]\rangle + \log\hat{\mathbb{E}}_P[\exp\left(X^T\delta X\right)],$$

where $\hat{\mathbb{E}}$ denotes an empirical mean, and $\delta$ is sparse. The second term, which is the only non-linear term, is reminiscent of normalization factors in graphical models. In this context, it is useful to recall the key ideas from high-dimensional sparse estimation theory (see [Neg+12]), which has served as a powerful generic tool. At a high-level, these results show that for a loss function $\mathcal{L}(\delta)$ paired with

a decomposable regulariser (such as an $\ell_1$ norm on $\delta$), if the loss function satisfies restricted strong convexity, namely, strong convexity only in a suitable descent error set, as characterised by the regulariser and the optimal value $\delta^*$, minimising the penalised empirical loss leads to a non-trivial estimation error bound. Leveraging these concepts of high-dimensional estimation, and exploiting sparsity, the sparse DCE works show that testing can be done in $O(\mathrm{poly}(s) \log p)$ samples (for any $P, Q$!), which is further much smaller than the number needed for $\mathbb{SL}$, a result which contradicts bounds we derive in this paper. The situation warrants further discussion.

From a technical perspective, the sample complexity gains of these methods arise from assuming law-dependent quantities to be constants. For example, [Liu+14; Liu+17] require that for $\|u\| \le \|\delta^*\|$, $\nabla^2 \mathcal{L}(\delta^* + u) \preccurlyeq \lambda_1 I$, and that for $S$ the support of $\delta^*$, the submatrix $(\nabla^2 \mathcal{L}(\delta^*))_{S,S} \succcurlyeq \lambda_2 I$, where $\lambda_1, \lambda_2$ are constants independent of $P, Q$. [FB16] removes the second condition, and shows that $\mathcal{L}$ has the $\lambda_2$-RSC property, where $\lambda_2$ is claimed to be independent of $P, Q$. In each case, sample costs increase with $\lambda_1$ and $\lambda_2^{-1}$. However, the assertion that $\lambda_1, \lambda_2$ are independent of $(P, Q)$ cannot hold in general – the only non-linear part in $\mathcal{L}$ is $\log \hat{\mathbb{E}}_P[\exp(X^T \delta X)]$, which clearly depends on $P$! This dependence also occurs if $P$ is known. Thus, the 'constants' $\lambda_1, \lambda_2$ are affected by the properties of $P$. More generically, the efficacy of sparse recovery techniques is questionable in this scenario. Since the data is essentially distinct across samples, and internally dependent, and since the sparse changes, $\delta^*$, and the underlying distributions interact, it is unclear if meaningful notions of design matrix that allow testing with sub-recovery sample costs can be developed.

Nevertheless, it is an interesting question to understand what additional assumptions on $P, Q$ or topological restrictions are useful in terms of benefiting from sparsity. Our results suggest that these conditions are stronger than typical incoherence conditions such as high temperatures, and further that the topological restrictions demand more than just 'simplicity' of the graphs.

**Other Methods.** [Cai+19] propose a method, whereby the parameters $\theta_P$ and $\theta_Q$ are only crudely estimated, and then tests using the biggest (normalised) deviations in the estimates as a statistic. The claims made in this paper are more modest, and do not show sample complexity below $n_{\mathrm{SL}}$. We point out, however, that $d$-dependent terms are treated as constants in this as well.

Much of the structural testing work studies Gaussian GMs instead of Ising (see the recent survey [Sho20]). We do not discuss these, but encourage the same careful examination of their assumptions.

**Other Information-Theoretic Approaches.** We adopted a similar information-theoretic viewpoint in our earlier work [GNS17; GNS18]. Of these, the former only considers the restricted case of $s = 1$ (very sparse changes), and the bounds in the latter are very inefficient. As such, the present paper is a significant extension and generalization of this perspective. Our bounds further improve the approximate recovery lower bounds of [SC16].

**Structural Testing Extensions.** A number of structural testing problems other than $\mathbb{GOF}$ have been pursued. For instance, [BN18] tests if the model is mean field or supported on a structured graph (sparse, etc.), [BN19] tests mean-field models against those on an expander, [CNL18] tests independence against presence of structure in high temperatures, [NL19] tests combinatorial properties of the underlying graph such as whether it has cycles, or the largest clique it contains (also see §E.2).

## 2   Problem Definitions and Notation

The zero external field Ising Model specifies a law on a $p$-dimensional random vector $X = (X_1, \ldots, X_p) \in \{\pm 1\}$, parametrised by a symmetric matrix $\theta$ with $0$ diagonal, of the form

$$ P_\theta(X = x) = \frac{\exp\left(\sum_{i<j} \theta_{ij} x_i x_j\right)}{Z(\theta)}, $$

where $Z(\theta)$ is called the partition function. Notice that given $X_j$ for all $j \in \partial i := \{j : \theta_{ij} \ne 0\}$, $X_i$ is conditionally independent of $X_{[1:p]-\{i\}-\partial i}$. Thus, the $\theta$ determine the local interactions of the model. With this intuition, one defines a simple, undirected graph $G(P_\theta) = ([1:p], E(P_\theta))$ with $E(P_\theta) = \{(i,j) : \theta_{ij} \ne 0\}$. This graph is called the *Markov network structure* of the Ising model, and $\theta$ can serves as a weighted adjacency matrix of $G(P_\theta)$. We often describe models by an unweighted graph, keeping weights implicit until required.

The model above can display very rich behaviour as $\theta$ changes, and this strongly affects all inference problems on Ising models. With this in mind, we make two explicit parametrisations to help us track how $\theta$ affects the sample complexity of various inference problems. The first of these is degree control - we assume that the degree of every node is $G(P), G(Q)$ is at most $d$. The second is weight control - we assume that if $\theta_{ij} \neq 0$, then $\alpha \leq |\theta_{ij}| \leq \beta$.

These are natural conditions: small weights are naturally difficult to detect, while large weights mask the nearby small-weight edges; degree control further sets up a local sparsity that tempers network effects in the models. The class of laws so obtained is denoted $\mathcal{I}_d(\alpha, \beta)$. We will usually work with a subclass $\mathcal{I} \subset \mathcal{I}_d$ which has *unique network structures* (i.e., for $P, Q \in \mathcal{I}, G(P) \neq G(Q)$). Note that we do not restrict $\alpha, \beta, d$ to have a particular behaviour - these are instead used as parametrisation to study how weights and degree affects sample complexity. In particular, they may vary with $p$ and each other. We do demand that $d \leq p^{1-c}$ for some constant $c > 0$, and that $p$ is large ($\gg 1$).

We let $\mathcal{G}$ be the set of all graphs on $p$ nodes, and $\mathcal{G}_d \subset \mathcal{G}$ be those with degree at most $d$. The symmetric difference of two graphs $G, H$ is denoted $G \triangle H$, which is a graph with edge set consisting of those edges that appear in exactly one of $G$ and $H$.

Lastly, we say that two Ising models are *s-separated* if their networks differ in at least $s$ edges. The 'anti-ball' $A_s(P) := \{Q \in \mathcal{I} : |G(Q) \triangle G(P)| \geq s\}$ is the set of $Q \in \mathcal{I}$ $s$-separated from $P$.

## 2.1 Problem Definitions

Below we define three structural inference problems: goodness-of-fit testing, error-of-fit identification, and approximate structure learning.

**Goodness-of-Fit Testing** Given $P$ and the dataset $X^n \sim Q^{\otimes n}$ where $Q \in \{P\} \cup A_s(P)$, we wish to distinguish between the case where the model is unchanged, $Q = P$, and the case where the network structure of the model differs in at least $s$ edges, $Q \in A_s(P)$. A goodness-of-fit test is a map $\Psi^{\mathrm{GoF}} : \mathcal{I} \times \mathcal{X}^n \to \{0, 1\}$. The $n$-sample risk is defined as

$$R^{\mathrm{GoF}}(n, s, \mathcal{I}) := \inf_{\Psi^{\mathrm{GoF}}} \sup_{P \in \mathcal{I}} \left\{ P^{\otimes n}(\Psi^{\mathrm{GoF}}(P, X^n) = 1) + \sup_{Q \in A_s(P)} Q^{\otimes n}(\Psi^{\mathrm{GoF}}(P, X^n) = 0) \right\}.$$

**Error-of-Fit Recovery** Given $P$ and the dataset $X^n \sim Q^{\otimes n}$ where $Q \in \{P\} \cup A_s(P)$ we wish to identify where the structures of $P$ and $Q$ differ, if they do. The error-of-fit learner is a graph-valued map $\Psi^{\mathrm{EoF}} : \mathcal{I} \times \mathcal{X}^n \to \mathcal{G}$. The $n$-sample risk is defined as

$$R^{\mathrm{EoF}}(n, s, \mathcal{I}) := \inf_{\Psi^{\mathrm{EoF}}} \sup_{P \in \mathcal{I}} \sup_{Q \in \{P\} \cup A_s(P)} Q^{\otimes n} \left( \left| \Psi^{\mathrm{EoF}}(P, X^n) \triangle (G(P) \triangle G(Q)) \right| \geq (s-1)/2 \right).$$

In words, $\Psi^{\mathrm{EoF}}$ attempts to recover $G(P) \triangle G(Q)$, and the risk penalises answers that get more than $(s-1)/2$ of the edges of this difference wrong. This problem is very similar to the following.

**s-Approximate Structure Learning** Given the dataset $X^n \sim Q^{\otimes n}$ we wish to determine the network structure of $Q$, with at most $s$ errors in the recovered structure. A structure learner is a graph-valued map $\Psi^{\mathrm{SL}} : \mathcal{X}^n \to \mathcal{G}$, and the risk of structure learning is

$$R^{\mathrm{SL}}(n, s, \mathcal{I}) := \inf_{\Psi^{\mathrm{SL}}} \sup_{Q \in \mathcal{I}} Q^{\otimes n}(|\Psi^{\mathrm{SL}}(X^n) \triangle G(P)| \geq s).$$

The sample complexity of the above problems is defined as the smallest $n$ necessary for the corresponding risk to be bounded above by $1/4$, i.e.

$$n_{\mathrm{GoF}}(s, \mathcal{I}) := \inf\{n : R^{\mathrm{GoF}}(n, s, \mathcal{I}) \leq 1/4\},$$

and similarly $n_{\mathrm{EoF}}$ and $n_{\mathrm{SL}}$ but with the risk lower bound of $1/8$.[1]

The above problems are listed in increasing order of difficulty, in that methods for $\mathbb{SL}$ yield methods for $\mathbb{EOF}$, which in turn solve $\mathbb{GOF}$. This is captured by the following statement, proved in §A.1.

**Proposition 1.** $n_{\mathrm{SL}}((s-1)/2, \mathcal{I}) \geq n_{\mathrm{EoF}}(s, \mathcal{I}) \geq n_{\mathrm{GoF}}(s, \mathcal{I})$.

Our main point of comparison with the literature on $\mathbb{SL}$ is the following result, which (mildly) extends [SW12, Thm 3a)] due to Santhanam & Wainwright. We leave the proof of this to Appx. A.2.

**Theorem 2.** *If $\mathcal{I} \subset \mathcal{I}_d(\alpha, \beta)$ has unique network structures, then for $s \le pd/2, \exists C \le 64$ such that*

$$n_{\mathrm{SL}}(s, \mathcal{I}) \le C \frac{d e^{2\beta d}}{\sinh^2(\alpha/4)} \left( 1 + \log \frac{p^2}{2s} + O(1/s) \right).$$

## 3  Lower Bounds for $\mathbb{GOF}$ and $\mathbb{EOF}$ over $\mathcal{I}_d(\alpha, \beta)$

This section states our results, and discusses our proof strategy, but proofs for all statements are left to §B. The bound are generally stated in a weaker form to ease presentation, but the complete results are described in §B. We begin by stating lower bounds for the case of $s = O(p)$. Throughout $500 > K > 1$ is a constant independent of all parameters.

**Theorem 3.** *If $20 \le d \le s \le p/K$, then there exists a $C > 0$ independent of $(s, p, d, \alpha, \beta)$ such that*

$$n_{\mathrm{GoF}}(s, \mathcal{I}) \ge C \max \left\{ \frac{e^{2\beta}}{\tanh^2 \alpha}, \frac{e^{2\beta(d-3)}}{d^2 \min(1, \alpha^2 d^4)} \right\} \log \left( 1 + C \frac{p}{s^2} \right)$$

$$n_{\mathrm{EoF}}(s, \mathcal{I}) \ge C \max \left\{ \frac{e^{2\beta}}{\tanh^2 \alpha}, \frac{e^{2\beta(d-3)}}{d^2 \min(1, \alpha^2 d^4)} \right\} \log \left( C \frac{p}{s} \right)$$

This statement is enough to make our generic point - for small $s$ (i.e., if $s \le p^{1/2-c}$ in $\mathbb{GOF}$ and if $s \le p^{1-c}$ in $\mathbb{EOF}$), the above bounds are uniformly within a $O(\mathrm{poly}(d))$ factor of the the upper bound on $n_{\mathrm{SL}}$ in Theorem 2. Notice also that the $\max$-terms are uniformly $\tilde{\Omega}(d^2)$ in the above - if $\beta d \ge 2 \log d$, then the second term in the max is $\Omega(d^2)$, while if smaller, the first term is $\Omega((d/\log d)^2)$ because $\alpha \le \beta$. Thus, over $\mathcal{I}_d$, the best possible sample complexity of $\mathbb{GOF}$ and $\mathbb{EOF}$ scales as $\tilde{\Omega}(d^2 \log p)$, and in particular cannot be generally $d$-independent.

Of course, graphs in $\mathcal{G}_d$ have upto $\sim pd$ edges, and so many more changes can be made. Towards this, we provide the following bound for $\mathbb{GOF}$. A similar result for $\mathbb{EOF}$ is discussed in §B.

**Theorem 4.** *If for some $\zeta > 0, s \le pd^{1-\zeta}/K$, and $d \ge 10$, then there exists a constant $C > 0$ independent of $(s, p, d, \alpha, \beta)$ such that*

1. *If $\alpha d^{1-\zeta} \le 1/32$ then $n_{\mathrm{GoF}} \ge C \dfrac{1}{d^{2-2\zeta} \alpha^2} \log \left( 1 + C \dfrac{pd^{3-3\zeta}}{s^2} \right)$.*

2. *If $\beta d \ge 4 \log(d-4)$ then $n_{\mathrm{GoF}} \ge C \dfrac{e^{2\beta d(1-d^{-\zeta})}}{d^2 \min(1, \alpha^2 d^4)} \log \left( 1 + C \dfrac{pd^{2-3\zeta}}{s^2} \right)$.*

Thm. 4 leaves a (small) gap, since as $\zeta \to 0$, $\alpha d^{1-\zeta} \le 1$ and $\beta d \ge 4 \log(d)$ do not completely cover all possibilities. Barring this gap, we again notice that for $s \ll \sqrt{pd^{1-\zeta}}$, $n_{\mathrm{GoF}}$ is separated from $n_{\mathrm{SL}}$ by at most a $\mathrm{poly}(d)$ factor. The first part of the above statement is derived using results of [CNL18]. For the limiting case of $\zeta = 0$, i.e. when $s$ is linear in $pd$, we recover similar bounds, but with the distinction that the $2\beta d$ in the exponent is replaced by a $\beta d$. See §B.

Finally, since often the interest in DCE lies in *very sparse* changes, we present the following -

**Theorem 5.** *If $s \le d$, then there exists a $C > 0$ independent of $(s, p, d, \alpha, \beta)$ such that*

$$n_{\mathrm{GoF}}(s, \mathcal{I}) \ge C \max \left\{ \frac{e^{2\beta}}{\tanh^2 \alpha}, \frac{e^{2\beta(d-1-2\sqrt{s})}}{d^6 \sinh^2(\alpha\sqrt{s})} \right\} \log \left( 1 + C \left( \frac{p}{s^2} \wedge \frac{p}{d} \right) \right)$$

$$n_{\mathrm{EoF}}(s, \mathcal{I}) \ge C \max \left\{ \frac{e^{2\beta}}{\tanh^2 \alpha}, \frac{e^{2\beta(d-1-2\sqrt{s})}}{d^6 \sinh^2(\alpha\sqrt{s})} \right\} \log \left( C \frac{p}{d} \right)$$

**Structure of the Bounds** Each of the bounds above can be viewed as of the form $(\mathrm{SNR})^{-1} \log(1 + f(p, s, d))$, where we call the premultiplying terms SNR since they naturally capture how much signal about the network structure of a law relative to its fluctuations is present in the samples. This SNR term in Thms. 3 and 5 is developed as a max of two terms. The first of these is effective in the

high temperature regime (where $\beta d$ is small), while the second takes over in the low temperature regime of large $\beta d$. Similarly, the first and second parts of Thm. 4 are high and low temperature settings, respectively, and have different SNR terms. The SNR in all of the above is within a $\mathrm{poly}(d)$ factor of the corresponding term in the upper bound for $n_{\mathrm{SL}}$.

The term $f(p, d, s)$ thus captures the hardness of testing/error localisation. For $\mathbb{EOF}$, as long as $s$ is small, this term takes the form $p^c$ for some $c$. Thus, generically, localising sparse changes is nearly as hard as approximate recovery. This is to be expected from the form of the $\mathbb{EOF}$ problem itself. More interestingly, for $\mathbb{GOF}$, these take the form $pd^c/s^2$. When $s \ll \sqrt{pd^c}$, this continues to look polynomial in $p$, and thus $\mathbb{GOF}$ is as hard as recovery. On the other hand, for $s$ much larger than this, $f$ becomes $o(1)$ as $p$ grows, and so $\log(1 + f) \approx f$ itself and the resulting bounds look like $(\mathrm{SNR})^{-1}pd^c/s^2$. In the setting of low temperatures with non-trivially large degree, these can still be super-polynomial in $p$, but relative to $n$ they are essentially vanishing.

Notice that in high temperatures ($\beta d \leq 1$), the bounds of Thms. 3 and 5 are only $O(d)$ away from $n_{\mathrm{SL}}$ for small $s$, fortifying our claim that $\mathbb{GOF}$ and $\mathbb{EOF}$ are not separated from $\mathbb{SL}$ in this setting.

**Counterpoint to Sparse DCE efforts** The above bounds, especially Thm. 5, show that for small $s$ $\mathbb{GOF}$ and $\mathbb{EOF}$ are as hard as recovery of $G(Q)$ itself. A possible critique of these bounds when considering DCE is that the DCE schemes demand that the changes are smaller than $s$, while our formulations only require the changes to have size at least $s$. To counter this, we point out that the constructions for Thms. 3, 4, and 5 make at most $2s$ changes when computing bounds for any $s$ (in fact, smaller edits lead to stronger bounds). Thus, the above results catergorically contradict the claim that a generic $O(\mathrm{poly}(s) \log p)$ bound that is $d$ independent and much smaller than $n_{\mathrm{SL}}$ can hold for DCE methods on $\mathcal{I}_d$. Since $\alpha, \beta, d$ are only parameters, and are not restricted in any way, this shows that the assumptions made for DCE cannot be reduced to some conditions on only $\alpha, \beta, d$, and further topological conditions must be implicit. In particular, these are stronger than typical incoherence conditions such as Dobrushin/high-temperature ($\beta d < 1$; e.g., [DDK17; GLP18]).

## 3.1 Proof Technique

The above bounds are shown via Le Cam's method with control on the $\chi^2$-divergence of a mixture of alternatives for $\mathbb{GOF}$, and via a Fano-type inequality for the $\chi^2$-divergence, due to Guntuboyina [Gun11] for $\mathbb{EOF}$. These methods allow us to argue the bounds above by explicit construction of distributions that are hard to distinguish. We briefly describe the technique used for $\mathbb{GOF}$ below.

**Definition** *A $s$-change ensemble in $\mathcal{I}$ is a distribution $P$ and a set of distributions $\mathcal{Q}$, denoted $(P, \mathcal{Q})$, such that $P \in \mathcal{I}, Q \subseteq \mathcal{I}$, and for every $Q \in \mathcal{Q}$, it holds that $|G(P) \triangle G(Q)| \geq s$.*

Each of the testing bounds we show will involve a mixture of $n$-fold distributions over a class of distributions. For succinctness, we define the following symbol for a set of distibutions $\mathcal{Q}$

$$\langle \mathcal{Q}^{\otimes n} \rangle := \frac{1}{|\mathcal{Q}|} \sum_{Q \in \mathcal{Q}} Q^{\otimes n}.$$

Le Cam's method (see e.g. [Yu97; IS12]) shows that if $(P, \mathcal{Q})$ is a $s$-change ensemble in $\mathcal{I}$, then

$$R^{\mathrm{GoF}}(n, s, \mathcal{I}) \geq 1 - \sqrt{\frac{1}{2} \log(1 + \chi^2(\langle \mathcal{Q}^{\otimes n} \rangle \| P^{\otimes n}))}.$$

As a consequence, if we find a change ensemble and an $n$ such that $1 + \chi^2(\langle \mathcal{Q}^{\otimes n} \rangle \| \mathcal{P}^{\otimes n}) \leq 3$, then we would have established that $n_{\mathrm{GoF}}(s, \mathcal{I}) \geq n$. So, our task is set up as constructing appropriate change ensembles for which the $\chi^2$-divergence is controllable.

Directly constructing such ensembles is difficult, essentially due to the combinatorial athletics involved in controlling the divergence. We instead proceed by constructing a pair of separated distributions $(P_0, Q_0)$ on a small number of nodes, and then 'lifting' the resulting bounds to the $p$ nodes via tensorisation - $P$ is contructed by collecting disconnected copies of $P_0$, while $\mathcal{Q}$ is constructed by changing some of the $P_0$ copies to $Q_0$. The process is summarised as follows.

**Lemma 6.** *(Lifting) Let $P_0$ and $Q_0$ be Ising models with degree $\leq d$ on $\nu \leq p/2$ nodes such that $|G(P_0) \triangle G(Q_0)| = \sigma$, and $\chi^2(Q_0^{\otimes n} \| P_0^{\otimes n}) \leq a_n$. Let $m := \lfloor p/\nu \rfloor$. For $t < m/16e$, there exists a $t\sigma$-change ensemble $(P, \mathcal{Q})$ over $p$ nodes such that $|\mathcal{Q}| = \binom{m}{t}$ and*

$$1 + \chi^2(\langle \mathcal{Q}^{\otimes n} \rangle \| P^{\otimes n}) \leq \exp\left(\frac{t^2}{m} a_n\right).$$

A similar argument is used for the $\mathbb{EOF}$ bounds, along with a similar lifting trick, discussed in §B. Due to the tensorisation of the $\chi^2$-divergence, we obtain results of the form $a_n \leq (1+\kappa)^n - 1$, where $\kappa$ depends on $(P_0, Q_0)$ but not $n$. Plugging this into the above with $t = \lceil s/\sigma \rceil$ yields

$$n_{\text{GoF}}(s, \mathcal{I}) \geq \frac{1}{\log(1+\kappa)} \log\left(1 + \frac{p\sigma^2}{8\nu s^2}\right).$$

Notice that this $\kappa$ is an SNR term, while $\log(1 + p\sigma^2/8\nu s^2)$ captures combinatorial effects.

The procedure thus calls for strong $\chi^2$ bounds for various choices of small graphs, or 'widgets'. We use two varieties of these - the first, 'star-type' widgets, are variations on a star graph. These allow direct calculations in general, and provide bounds that extend to the high-temperature regime. The second variety is the 'clique-type' widgets, that are variations on a clique, and provide low-temperature obstructions. Classical Curie-Weiss analysis shows that cliques tend to 'freeze' - for Ising models on a $k$-clique with uniform weight $\lambda$, the probability mass concentrates on the set $\{(1)^{\otimes k}, (-1)^{\otimes k}\}$ w.p. roughly $1 - e^{-\Theta(\lambda k)}$. The clique-type obstructions implicitly argue that this effect is very robust.

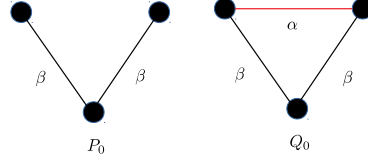

Figure 1: Graphs used to construct high-temperature obstructions. Labels indicate edge-weight, and the red edge is added in $Q_0$.

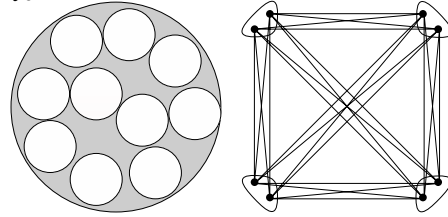

Figure 2: Two views of Emmentaler cliques. Left: the base clique is the large grey circle, uncoloured circles represent the groups with no edges within (this is $d, \ell \gg 1$, $d+1/\ell+1 = 10$); Right: Emmentaler as the graph $K_{\ell+1, \ell+1, \ldots, \ell+1}$ ($d = 7, \ell = 1$).

The particular graphs used to argue the high temperature bounds in Thms. 3,5 are a 'V' versus a triangle as seen in Fig. 1, while in Thm. 4 the empty graph is compared to a $d^{1-\zeta}$-clique. The low temperature obstructions of Thms. 3,4 compare a full $d+1$-clique as $P_0$ to an 'Emmentaler' clique (Fig. 2). These are constructed by dividing the $d+1$ nodes into groups of size $\ell+1$, and removing the $\ell+1$-subclique within each group. The graph can thus be seen either as a clique with many large 'holes' - corresponding to the deleted subcliques - which inspires the name, or as the complete $d+1/\ell+1$-partite graph on $d+1$ nodes. Notice that in the Emmentaler clique we have deleted $\approx d\ell/2$ edges. We will show in §D that this is still hard to distinguish from the full clique for $\ell \sim d/10$ - a deletion of $\Omega(d^2)$ edges!

**On Tightness** Prima facie the above bounds suggest that one may find sample efficient schemes in, say, $\mathbb{GOF}$ for $s \gg \sqrt{pd}$. However, it is our opinion that these bounds are actually loose. Particularly, while the SNR terms are relatively tight, the behaviour of $f(p, d, s)$ is not. To justify this opinion, consider the setting of forest-structured graphs. By the same techniques, we show a similar bound with $f = p/s^2$ for $\mathbb{GOF}$ in forests in §4.1 - this is the best possible by the methods employed. For $s \gg \sqrt{p}$, the resulting overall lower bound is the trivial $n \geq 1$ unless $\alpha \leq (p/s^2)^{1/2}$. On the other hand, [DDK19, Thm. 14] can be adapted to show a lower bound for forests of $\Omega(\alpha^{-2} \wedge \alpha^{-4}/p)$ for the particular case of $s = p/2$, which is non-trivial for all $\alpha \lesssim p^{-1/4}$. Our results trivialise for $\alpha \gtrsim p^{-1/2}$ for this case, demonstrating looseness.

The reason for this gap lies in the lifting trick used to show these bounds. The tensorisation step involved in this constricts the set of 'alternates' one can consider, thus diminishing $f$. More concretely - there are about $p^2 - pd/2$ potential ways to add an edge (and $O(pd)$ to delete an edge), while the lifting process as implemented here restricts these to at most $O(pd)$. It is important to recognize this lossiness, particularly since *most* lower bounds, for both testing and recovery, proceed via a similar trick, e.g. [SW12; Tan+14; SC16; GNS17; NL19; CNL18]. [DDK19, Thm. 14] is the only exception we know of. We conjecture that for $\mathbb{GOF}$ in $\mathcal{I}_d$, $f$ should behave like $p^2/s^2$, while for $\mathbb{EOF}$, it should behave like $p^2/s$. Note that for $\mathbb{GOF}$, since $s$ can be as big as $pd$, this indicates that one should look for sample-efficient achievability schema in the setting of $s > pd^c$.

However, for simpler settings this technique *can* recover tight bounds. For instance, §4.1 presents a matching upper bound for testing of edge-*deletion* in a forest. Notice that in this case there are only $O(p)$ possible ways to edit. This raises the further question of if the same effect extends to $\mathcal{I}_d$, i.e., can deletion of edges in $\mathcal{I}_d$ be tested with $O(1 \vee e^{2\beta d}\alpha^{-2}(pd/s^2))$ samples when $s \gg \sqrt{pd}$? §4.2 offers initial results in this direction in the high temperature regime.

# 4 Testing Edge Deletions

Continuing on the theme that concluded our discussion of the tightness of our lower bounds, we study the testing of edge deletions in two classes of Ising models - forests, and high-temperature ferromagnets - with the aim demonstrating natural settings in which the sample complexity of $\mathbb{GOF}$ testing of Ising models is provably separated from that of the corresponding recovery problem.

In the deletion setting, we consider the same problems as in §2, but with the additional constraint that if $Q \neq P$, then $G(Q) \subset G(P)$, that is, the network structures of alternates can be obtained by dropping some edges in that of the null. For a class of Ising models $\mathcal{J}$, we thus define

$$R^{\mathrm{GoF,del}}(n, s, \mathcal{J}) = \inf_{\Psi} \sup_{P \in \mathcal{J}} P^{\otimes n}(\Psi(P, X^n) = 1) + \sup_{\substack{Q \in A_s(P) \cap \mathcal{J} \\ G(Q) \subset G(P)}} Q^{\otimes n}(\Psi(P, X^n) = 1),$$

and, analogously define $R_{\mathrm{EoF,del}}$, and the sample complexities $n_{\mathrm{GoF,del}}(s, \mathcal{J})$ and $n_{\mathrm{EoF,del}}(s, \mathcal{J})$.

We will look at testing deletions for two choices of $\mathcal{J}$ which both have uniform edge weights

• **Forest-Structured Models** ($\mathcal{F}(\alpha)$) are Ising models with uniform weight $\alpha$ such that their network structure is a forest (i.e., has no cycles).
• **High-Temperature Ferromagnets** ($\mathcal{H}_d^\eta(\alpha)$) are models with max degree at most $d$, uniform *positive* edge weights $\alpha$, and further such that there is an $\eta < 1$ such that $\alpha d \leq \eta$.

We note that while our motivation for the study of the above is technical, both of these subclasses of models have been utilised in practice, and indeed are the subclasses of $\mathcal{I}_d$ that are best understood.

## 4.1 Testing Deletions in Forests

Forest-structured Ising models are known to be tractable, and have thus long served as the first setting to explore when trying to establish achievability statements. We show a tight characterisation of the sample complexity of testing deletions in forests for large changes, and also demonstrate the separation from the corresponding $\mathbb{EOF}$ (and thus also $\mathbb{SL}$) problem. In addition, we also show that for the restricted subclass of trees, essentially the same characterisation follows for *arbitrary* changes (i.e., not just deletions), and that the methods support some amount of tolerance directly. We begin with the main result for testing deletions in forests (all proofs are in §C.1).

**Theorem 7.** *There exists a constant $C$ independent of $(s, p, \alpha)$ such that the sample complexity of $\mathbb{GOF}$ testing of forest-structured Ising models against deletions is bounded as*

$$n_{\mathrm{GoF,del}}(s, \mathcal{F}(\alpha)) \leq C \max\left\{1, \frac{1}{\sinh^2(\alpha)} \frac{p}{s^2}\right\}.$$

*Conversely, for $s \leq p/32e$, there exists a constant $C'$ independent of $(s, p, \alpha)$, such that*

$$n_{\mathrm{GoF,del}}(s, \mathcal{F}(\alpha)) \geq \max\left\{1, \frac{1}{C'} \frac{1}{\sinh^2 \alpha} \log\left(1 + \frac{p}{C's^2}\right)\right\},$$

$$n_{\mathrm{EoF,del}}(s, \mathcal{F}(\alpha)) \geq \frac{1}{C' \sinh^2 \alpha} \log\left(\frac{p}{C's}\right).$$

The upper bound is constructed by using the simple global statistic $\mathscr{T}_P = \sum_{(i,j) \in G(P)} X_i X_j$, averaged across the samples. Again, the behaviour of the lower bound shifts as $s$ crosses $\sqrt{p}$ - for larger $s$, it scales as $1 \vee \sinh^{-2}(\alpha) p/s^2$, while for much smaller $s$ it is $1 \vee \sinh^{-2}(\alpha) \log p$. Further, for large changes, the lower bound is matched, up to constants, by the achievability statement above. For the smaller case, the same holds in the restricted setting of $\alpha < 1$, since exact recovery in $\mathcal{F}(\alpha)$ only needs $\tanh^{-2}(\alpha) \log p$ samples (Chow-Liu algorithm, as analysed in [BK16]).[2] Finally, the $\mathbb{EOF}$ lower bound (which is also tight for $\alpha < 1$, show that the sample complexity of $\mathbb{GOF}$ is separated from error of fit (and thus $\mathbb{SL}$) for large changes.

Fig. 3 illustrates Thm. 7 via a simulation for testing deletions in a binary tree (for $p = 127, \alpha = 0.1$), showing excellent agreement. In particular, observe the sharp drop in samples needed at $s = 21 \approx 2\sqrt{p}$ versus at $s < \sqrt{p} \approx 11$. We note that $\mathbb{SL}$-based testing fails for all $s \leq 60$ for this setting even with 1500 samples (Fig. 4 in §C.3), which is far beyond the scale of Fig. 3. See §C.3 for details.

**Testing arbitrary changes in trees** The statistic $\mathcal{T}$ is good at detecting deletions in edges, but is insensitive to edge additions, which prevents it from being effective in general for forests. However, if the forest-models $P$ and $Q$ are restricted to have the same *number of edges*, then $\mathcal{T}$ should retain power, since any change of $s$ edges must delete $s/2$ edges. This, of course, naturally occurs for trees! Let $\mathcal{T}(\alpha) \subset \mathcal{F}(\alpha)$ denote tree-structured Ising models.

**Theorem 8.** *There exists a $C$ independent of $(p, s, \alpha)$ s.t.*

$$n_{\text{GoF}}(s, \mathcal{T}(\alpha)) \leq C \max\left(1, \frac{1}{(1 - \tanh(\alpha))^2 \sinh^2(\alpha)} \frac{p}{s^2}\right).$$

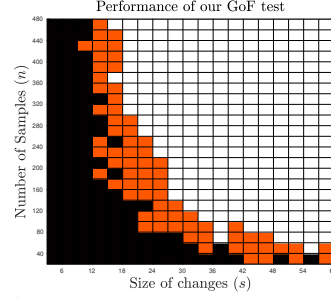

Figure 3: Testing deletions in binary trees for $p = 127, \alpha = 0.1$. Entries are coloured black if risk is $> 0.35$, white if $< 0.15$, and orange otherwise.

**Tolerant Testing** The achievability results of Thm.s 7,8 can be made 'tolerant' without much effort (see §C.1.3). 'Tolerance' here refers to updating the task to separate models that are $\varepsilon s$-close to $P$ from those that are $s$-far from it. The key point here is that for $\tau = \tanh(\alpha)$, changing $\varepsilon s$ edges reduces the mean of $\mathcal{T}_P$ by at most $\varepsilon s \tau$ in both cases, while changing $\geq s$ edges reduces it by at least $s\tau$ for forest deletion, and $s\tau(1-\tau)/2$ for arbitrary changes in trees. Thus, tolerant testing has a blow up in sample costs of $(1 - \varepsilon)^{-2}$ for forest deletions, and of $O((1 - 2\varepsilon - \tau)^{-2})$ for trees (if $\varepsilon < 1-\tau/2$). This should be contrasted with statistical distance based formulations of testing, for which tolerant testing is a subtle question, and, at least in unstructured settings, requires using different divergences to define closeness and farness in order to show gains beyond learning [DKW18].

### 4.2   Testing Deletions in High-Temperature Ferromagnets

Testing deletions in ferromagnets is amenable due to two technical properties of the statistic $\mathcal{T}_P = \sum_{(i,j) \in G(P)} X_i X_j$. The first of these is that due to the ferromagneticity, deleting an edge can only reduce the correlations between the values that the variables take. Coupling this fact with a structural result that is derived using [SW12, Lemma 6] yields that if $G(Q) \subset G(P)$ and $|G(P) \triangle G(Q)| \geq s$, then $\mathbb{E}_P[\mathcal{T}_P] - \mathbb{E}_Q[\mathcal{T}_P] \gtrsim s\alpha$. The second technical property is that bilinear functions of the variables, such as $\mathcal{T}_P$, exhibit concentration in high-temperature Ising models. In particular, using the Hoeffding-type concentration of [Ada+19, Ex. 2.5], $\mathcal{T}_P$ concentrates at the scale $O(\sqrt{pd})$ around its mean for all high-temperature ferromagnets. With means separated, and variances controlled, we can offer the following upper bound on the sample complexity, while the converse is derived using techniques of previous sections. See §C.2 for proofs.

**Theorem 9.** *There exists a constant $C_\eta$ depending only on $\eta$ and not on $(s, p, d, \alpha)$ such that*

$$n_{\text{GoF,del}}(s\mathcal{H}_d^\eta(\alpha)) \leq C_\eta \left(\frac{pd}{\alpha^2 s^2} \vee 1\right).$$

*Conversely, there exists a $c < 1$ independent of $(s, p, d, \alpha)$ such that if $\eta \leq 1/16, s \leq cpd$ then*

$$n_{\text{GoF,del}}(s, \mathcal{H}_d^\eta(\alpha)) \geq \frac{c}{\alpha^2 d^2} \log\left(1 + \frac{cpd^3}{s^2}\right) \ \& \ n_{\text{EoF,del}}(s, \mathcal{H}_d^\eta(\alpha)) \geq \frac{c}{\alpha^2 d^2} \log\left(1 + \frac{cpd}{s}\right)$$

Unlike in Thm. 7, the lower bounds above are not very clean, and so our characterisation of the sample complexity is not tight. Nevertheless, we once again observe a clear separation between sample complexities of $\mathbb{GOF}$ and of $\mathbb{EOF}$ and a fortiori that of $\mathbb{SL}$. Concretely, our achievability upper bound and the $\mathbb{EOF}$ lower bound show that for $s > \sqrt{pd^3}$, the sample complexity of testing deletions is far below that of structure learning in this class. Further, our testing lower bound tightly characterises the sample complexity for $s \geq \sqrt{pd^3}$.

As an aside, note that unlike in the forest setting, it is not clear if $\mathcal{T}$ is generically sensitive to edge deletions, since network effects due to cycles in a graph can bump up correlation even for deleted edges. However, we strongly suspect that a similar effect does hold in this setting, raising another open question - can testing of changes in the subclass of $\mathcal{H}_d^\eta$ with a fixed number of edges be performed with $O(\alpha^{-2} pd/s^2)$ samples for large $s$? A similar open question arises for tolerant testing, which requires us to show that small changes do not alter the mean of $\mathcal{T}$ too much.

**Broader Impact**    Our work is theoretical. It primarily investigates the limits of finding changes in network structure in settings that are amenable to graphical models. Secondarily, it identifies regimes in which to focus algorithmic design of tests of network structure, and gaps in the characterisation of existing algorithmic approaches to the same. As such, the immediate impact it has is only on theoretical explorations.

**Acknowledgements**    AG would like to thank Bodhi Vani and Anil Kag for discussions that helped with the simulations described in §C.3, on which Figure 3 is based.

**Funding Disclosure**    This work was supported by the National Science Foundation grants CCF-2007350 (VS), DMS-2022446 (VS), CCF-1955981 (VS and BN) and CCF-1618800 (AG and BN). AG was funded in part by VS's data science faculty fellowship from the Rafik B. Hariri Institute at Boston University. We declare that we have no competing interests.

## Footnotes

[1]1/4 is convenient for bounds for $\mathbb{GOF}$, but any risk smaller than 1 is of interest, and can be boosted to arbitrary accuracy by repeating trials and majority. For $\mathbb{EOF}, \mathbb{SL}$ we use 1/8 for ease of showing Prop. 1.

[2]While the $\alpha < 1$ regime is certainly more relevant in practice, it is an open question whether for larger $\alpha$, and for small $s$, the correct SNR behaviour is $\sinh^{-2}$ or $\tanh^{-2}$ in testing.

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
