[Supplementary Material]

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

[3]note however that exactly one of $\theta_{ab}^P$ and $\theta_{ab}^Q$ is zero, since $(a, b)$ lies in one but not the other graph.

[4]The expression $2\alpha d \max_{u \in \{a, b\}, v \in V} |\mu_{uv} - \mu'_{uv}|$ already accounts for the extra term we add, since it allows us to take $u = a, v = b$.

[5]Instead of the Frobenius norm $\|A\|_F$, the bound of [Ada+19] features the Hilbert-Schmidt norm of $A$. These are the same thing for finite dimensional operators.

[6]This effect is linked to the concentration of the Ising model on the clique we mentioned before. Notice that the probability of a uniform state is as $1 - \exp\left(-\Omega(\beta d)\right)$. For this to be appreciable, i.e., at least polynomially close to 1, a condition like $\beta d = \Omega(\log d)$ is in fact necessary.

[7]It should be noted that this analogy is flawed - while the notions of being close are indeed similar, the notion of being far from a model is significantly different under the two formulations. The main text mentions an example illustrating this - if a small group of disconnected nodes is bunched into a clique, a large statistical change is induced due to the marked difference in the marginal law of this group, but the structural change is tiny. Of course, being close and far are ultimately related concepts, and some shadow of this effect must be cast on the closeness argument we have just presented.

[8] i.e. of cardinality 0 or 1. More precise characterisation can be obtained by casework on the number of roots of $\tau'$.

[9]Unlike in §F.2.1, we include the factor due to $\eta$ in the normalisation. This does not affect the further calculations since these factors cancel in the expression for $W$ below. More importantly, the normalisation includes a factor of $e^{\lambda/2((d+1)^2 - (d+1))}$ instead of $e^{\lambda/2(d^2-d)}$. While the latter lent the formulae in the $\ell = 2$ case of the previous section a pleasant symmetry, the former yields more convenient expressions when dealing with $\ell$ abstractly. Due to this, the terms are further reduced by a common factor of $e^{\lambda d}$. We highlight this here because of the cosmetic differences arising from these changes—for instance, the leading term in $\widetilde{Z}_\ell$ is just $S_1$ instead of $e^{\lambda d - \eta} S_1$ as in the §F.2.1—which may irk the careful reader at first glance.

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

# Appendices

## A   Appendix to §2

### A.1   Proof of Ordering of Sample Complexities

The proposition is argued by direct reductions showing how a solver of a harder problem can be used to solve a simpler problem. The main feature of the definitions that allows this is that the risks of $\mathbb{SL}$ and $\mathbb{EOF}$ are defined in terms of a probability of error.

*Proof of Proposition 1.*

*Reducing EoF to SL*: Suppose we have a $(s-1/2)$-approximate structure learner with risk $\delta$ that uses $n$ samples. Then we can construct the following $\mathbb{EOF}$ estimator with the same sample costs. Take a dataset from $Q^{\otimes n}$, and pass it to the structure learner. With probability at least $1 - \delta$, this gives a graph $\widehat{G}$ that is at most $\lfloor s/2 \rfloor$-separated from $G(Q)$. Now compute $G(P) \triangle \widehat{G}$ ($G(P)$ is determined because $P$ is given to the $\mathbb{EOF}$ tester). By the triangle inequality applied to the adjacency matrices of the graphs under the Hamming metric, this identifies $G(P) \triangle G(Q)$ up to an error of $(s-1)/2$, and so, the EoF risk incurred is also $\delta$. Taking $\delta = 1/8$ concludes the argument.

*Reducing GoF to EoF*: Suppose we have a $s$-EoF solver that uses $n$ samples with risk $\delta$. Again, take a dataset from $Q^{\otimes n}$, and pass it to the EoF solver, along with $P$. With probability at least $1 - \delta$, this yields a graph $\widehat{G}$ such that $|\widehat{G} \triangle (G(P) \triangle G(Q))| \leq (s-1)/2$. But then, if $G(Q) = G(P)$, $\widehat{G}$ can have at most $(s-1)/2$ edges, while if $|G(P) \triangle G(Q)| \geq s$, then $\widehat{G}$ must have at least $(s+1)/2$ edges. Thus, thresholding on the basis of the number of edges in $\widehat{G}$ produces a GoF tester with both null and alternate risk controlled by $\delta$, or total risk $2\delta$. Taking $\delta = 1/8$ then finishes the argument.   $\square$

### A.2   Proof of Upper Bound on $n_{\mathrm{SL}}$

This proof is essentially constructed by slightly improving upon the proof of [SW12, Thm 3a)] due to Santhanam & Wainwright, which analyses the maximum likelihood scheme. We use notation from that paper below.

*Proof of Theorem 2.* [SW12] shows, in Lemmas 3 and 4, that if the data is drawn from an Ising model $P \in \mathcal{I}_d$, and $Q \in \mathcal{I}_d$ is such that $G(P) \triangle G(Q) = \ell$, then

$$P^{\otimes n}(\mathscr{L}(P) \leq \mathscr{L}(Q)) \leq \exp\left(-n\ell\kappa/8d\right),$$

where $\mathscr{L}(P)$ denotes the likelihood of $P$, i.e. if the samples are denoted $\{X^{(k)}\}_{k \in [1:n]}$, then $\mathscr{L}(P) = \prod_{k=1}^{n} P(X^{(k)})$, and

$$\kappa = (3e^{2\beta d} + 1)^{-1} \sinh^2(\alpha/4) \geq \frac{\sinh^2(\alpha/2)}{4e^{2\beta d}}.$$

Now, for the max-likelihood scheme to make an error in approximate recovery, it must make an error of at least $s$ - i.e., an error occurs only if $\mathscr{L}(Q) \geq \mathscr{L}(P)$ for some $Q$ with $G(Q) \triangle G(P) \geq s$. Union bounding this as Pg. 4129 of [SW12], we may control this as

$$\begin{aligned}
P(\mathrm{err}) &\leq \sum_{\ell=s}^{pd} \binom{\binom{p}{2}}{\ell} \exp\left(-n\ell\kappa/8d\right) \\
&\leq \sum_{\ell=s}^{pd} \exp\left(\ell\left(\log\frac{ep^2}{2\ell} - n\kappa/8d\right)\right) \\
&\leq \sum_{\ell=s}^{pd} \exp\left(\ell\left(\log\frac{ep^2}{2s} - n\kappa/8d\right)\right).
\end{aligned}$$

Now, if $n\kappa/8d \geq 2\log{}^{ep^2}/_{2s} = 2\log p^2/s + 2(1 - \log(2))$, and if $\exp(-ns\kappa/8d) \leq {}^1/_2$ then the above is bounded as $2\exp(-ns\kappa/8d)$, which can be driven lower than any $\delta$ by increasing $n$ by an $O(s^{-1}\log(2/\delta))$ additive factor. It follows that

$$n_{\mathrm{SL}}(s, \mathcal{I}) \leq \frac{16d}{\kappa}\left(\log\frac{p^2}{s} + 2 + O(1/s)\right),$$

and the claim follows by expanding out the value of $\kappa$. $\qquad\square$

# B  Appendix to §3

## B.1  Expanded Proof Technique

This section expands upon §3.1 in the main text, including a treatment of the method used for $\mathbb{EOF}$ lower bounds, giving an expanded version of Lemma 6, and a theorem collating the resulting method to construct bounds. Some of the text from §3.1 is repeated for the sake of flow of the presentation.

As discussed previously, the proofs proceed by explicitly constructing distributions with differing network structures that are statistically hard to distinguish. In particular, we measure hardness by the $\chi^2$-divergence. We begin with some notation.

**Definition** *A s-change ensemble in $\mathcal{I}$ is a distribution $P$ and a set of distributions $\mathcal{Q}$, denoted $(P, \mathcal{Q})$, such that $P \in \mathcal{I}, Q \subseteq \mathcal{I}$, and for every $Q \in \mathcal{Q}$, it holds that $|G(P)\triangle G(Q)| \geq s$.*

Each of the testing bounds we show will involve a mixture of $n$-fold distributions over a class of distributions. For succinctness, we define the following symbol.

**Definition** *For a set of distributions $\mathcal{Q}$ and a natural number $n$, we define the mixture*

$$\langle\mathcal{Q}^{\otimes n}\rangle := \frac{1}{|\mathcal{Q}|}\sum_{Q\in\mathcal{Q}} Q^{\otimes n}.$$

Consider the case of $\mathbb{GOF}$ testing, with the known distribution $P$. Suppose we provide the tester with the additional information that the dataset is drawn either from $P$, or from a distribution picked uniformly at random from $\mathcal{Q}$, where $(P, \mathcal{Q})$ for a $s$-change ensemble. Clearly, the Bayes risk suffered by any tester with this side information must be lower than the minimax risk of $\mathbb{GOF}$ testing. The advantage of this formulation is that the risks of these tests with the side information can be lower bounded by standard techniques - basically the Neyman-Pearson Lemma. The following generic bound, which is Le Cam's two point method [Yu97; IS12] captures this.

**Lemma 10.** *(Le Cam's Method)*

$$R^{\mathrm{GoF}}(n, s, \mathcal{I}) \geq \sup_{(P,\mathcal{Q})} 1 - d_{\mathrm{TV}}(\langle\mathcal{Q}^{\otimes n}\rangle, P^{\otimes n}) \geq \sup_{(P,\mathcal{Q})} 1 - \sqrt{\frac{1}{2}\log(1 + \chi^2(\langle\mathcal{Q}^{\otimes n}\rangle\|P^{\otimes n}))},$$

*where the supremum is over s-change ensembles in $\mathcal{I}$.*

Above, $\chi^2(\cdot\|\cdot)$ is the $\chi^2$-divergence, which is defined for distributions $P, Q$ as follows

$$\chi^2(Q\|P) := \begin{cases} \mathbb{E}_P\left[\left(\frac{\mathrm{d}Q}{\mathrm{d}P}\right)^2\right] - 1 & \text{if } Q \ll P \\ \infty & \text{if } Q \not\ll P \end{cases}.$$

Note that generally the method is only stated as the first bound, and the second is a generic bound on the total variation divergence which follows from Pinsker's inequality and the monotonicity of Rényi divergences. The $\chi^2$-divergence is invoked becuase it yields a twofold advantage in that it both tensorises well, and behaves well under mixtures such as $\langle\mathcal{Q}^{\otimes n}\rangle$ above.

For the $\mathbb{EOF}$ bounds, more care is needed. Recall that the $\mathbb{EOF}$ problem only requires errors smaller than $s/2$. To address this, we introduce the following.

**Definition** *An $(s', s)$-packing change ensemble is an $s$-change ensemble $(P, \mathcal{Q})$ such that $\mathcal{Q}$ is an $s'$-packing under the Hamming metric on network structures, that is, for every $Q, Q' \in \mathcal{Q}, |G(Q)\triangle G(Q')| \geq s'$.*

Clearly, if one can solve the $\mathbb{EOF}$ problem, one can exactly recover the structures in a $(s/2, s)$-packing change ensemble. Thus, the following lower bound of Guntuboyina is applicable.

**Lemma 11.** *[Gun11, Example II.5]*

$$R^{\mathrm{EoF}}(n, s, \mathcal{I}) \geq \sup_{(P, \mathcal{Q})} 1 - \frac{1}{|\mathcal{Q}|} - \sqrt{\frac{\sum_{Q \in \mathcal{Q}} \chi^2(Q\|P)}{|\mathcal{Q}|^2}},$$

*where the supremum is taken over $(s/2, s)$-packing change ensembles in $\mathcal{I}$.*

Note that [Gun11] shows a number of lower bounds of the above form. We use the $\chi^2$-divergence here primarily for parsimony of effort, in that the bounds on $\chi^2$-divergences we construct for the $\mathbb{GOF}$ setting can easily extended to the $\mathbb{EOF}$ case via the above.

Our task is now greatly simplified - we merely have to construct change ensembles such that $|\mathcal{Q}|$ is large, and $\chi^2(Q\|P)$ is small for every $Q \in P$. Since it is difficult to directly construct large degree bounded graphs with tractable distributions, we will instead provide constructions on a small number of nodes, and lift these up to the whole $p$ nodes by the following lemma.

**Lemma 12.** *(Lifting) Let $P_0$ and $Q_0$ be Ising models with degree $\leq d$ on $\nu \leq p$ nodes such that $|G(P_0) \triangle G(Q_0)| = \sigma$, and $\chi^2(Q_0^{\otimes n}\|P_0^{\otimes n}) \leq a_n$. Let $m := \lfloor p/\nu \rfloor$. For $1 \leq t < m/16e$, there exists a $t\sigma$-change ensemble $(P, \mathcal{Q})$ over $p$ nodes such that $|\mathcal{Q}| = \binom{m}{t}$ and*

$$\chi^2(\langle \mathcal{Q}^{\otimes n} \rangle \| P^{\otimes n}) \leq \frac{1}{\binom{m}{t}} \sum_{k=0}^{t} \binom{t}{k} \binom{m-t}{t-k} ((1 + a_n)^k - 1) \leq \exp\left(\frac{t^2}{m} a_n\right) - 1.$$

*Further, there exists a $(t\sigma/2, t\sigma)$-packing change ensemble $(P, \widetilde{\mathcal{Q}})$ over $p$ nodes such that*

$$|\widetilde{\mathcal{Q}}| \geq \frac{2}{t} \left(\frac{m}{8et}\right)^{t/2}$$

*and*

$$\forall Q \in \widetilde{\mathcal{Q}}, \chi^2(Q^{\otimes n}\|P^{\otimes n}) \leq (1 + a_n)^t - 1.$$

We note that the proof of the above lemma constructs explicit change ensembles. We will abuse terminology and refer to *the* change ensemble or *the* packing change ensemble of Lemma 12.

The above Lemma requires control on $n$-fold products of two distributions. However, since the $\chi^2$-divergence is conducive to tensorisation, control for $n = 1$ is usually sufficient. The statement below captures this fact and gives an end to end lower bound on this basis. The statement amounts to collating the various facts described in this section.

**Theorem 13.** *Let $P_0$ and $Q_0$ be as in Lemma 12. Suppose further that $\chi^2(Q_0\|P_0) \leq \kappa$. Then for $1 \leq t < m/16e$, where $m = \lfloor p/\nu \rfloor$,*

$$n_{\mathrm{GoF}}(t\sigma, \mathcal{I}_d) \geq \frac{1}{2\log(1 + \kappa)} \log\left(1 + \frac{m}{t^2}\right),$$

$$n_{\mathrm{EoF}}(t\sigma, \mathcal{I}_d) \geq \frac{1}{2\log(1 + \kappa)} \log\left(\frac{m}{4000t}\right).$$

The 4000 in the above can be improved under mild assumptions, such as if $t \geq 8$, but we do not pursue this further. We conclude this section with proofs of the main claims above.

### B.1.1 Proof of Lifting Lemma

*Proof of Lemma 6.* Let $G_0, H_0$ be the network structures underlying $P_0, Q_0$, and $A_0, B_0$ be the weight matrices of $G_0, H_0$. Recall that these are graphs on $\nu$ nodes. Partition $[1 : p]$ into $m + 1$ pieces $(\pi_1, \pi_2, \ldots, \pi_m) = ([1 : \nu], [\nu + 1 : 2\nu], \ldots [(m-1)\nu + 1 : m\nu])$ and $\pi_{m+1} = [m\nu + 1 : p]$, the last one being possibly empty. We may place a copy of $G_0$ on each of the first $m$ parts, and leave the final graph disconnected to obtain a graph $G$ with the block diagonal weight matrix $\mathrm{diag}(A_0, A_0, \ldots, A_0, 0)$. We let $P$ be the Ising model on $G$. For any vector $\mathbf{v} \in \{0, 1\}^m$ of weight

$t$, let $Q_{\mathbf{v}}$ be the graph which places a copy of $B_0$ on $\pi_i$ for all $i : \mathbf{v}_i = 1$, and $A_0$ as before otherwise. Note the block independence across parts of $\pi$ induced by this. Concretely, we have

$$P(X = x) = \prod_{i=1}^{m} P_0(X_{\pi_i} = x_{\pi_i}) \cdot 2^{-|\pi_{m+1}|},$$

$$Q_{\mathbf{v}}(X = x) = P(X = x) \cdot \prod_{i:\mathbf{v}_i=1} \frac{Q_0(X_{\pi_i} = x_{\pi_i})}{P_0(X_{\pi_i} = x_{\pi_i})}.$$

Now, let $\mathcal{V}_t$ be the $t$-weighted section of the cube $\{0,1\}^m$, and $\mathcal{V}_t'$ be a maximal $t/2$ packing of $\mathcal{V}_t$.

We let $\mathcal{Q} := \{Q_{\mathbf{v}}, \mathbf{v} \in \mathcal{V}_t\}$ and $\mathcal{Q}' := \{Q_{\mathbf{v}}, \mathbf{v} \in \mathcal{V}_t'\}$. Since $(P_0, Q_0)$ had symmetric difference $\sigma$, and since we introduce $t$ differences of this form in $\mathcal{Q}$, $(P, \mathcal{Q})$ forms a $t\sigma$-change ensemble. Further, $\mathcal{Q}'$ inherits the packing structure of $\mathcal{V}_t'$, $(P, \mathcal{Q}')$ forms a $(t\sigma/2, t\sigma)$-packing change ensemble. Next note that $|\mathcal{Q}| = \binom{m}{t}$ trivially. Further, since $|\mathcal{Q}'| = |\mathcal{V}_t|$, it suffices to lower bound the latter to show that $\mathcal{Q}$ is as big as claimed. Since $\mathcal{V}_t'$ is maximal, its cardinality must exceed the $t/2$-covering number of the $t$-section of the cube. But then, by a volume argument,

$$|\mathcal{V}_t'| \geq \frac{\binom{m}{t}}{\sum_{k=0}^{t/2} \binom{t}{k}\binom{m-t}{k}} \geq \frac{\binom{m}{t}}{(t/2)2^t\binom{m}{t/2}} \geq \frac{2}{t}\left(\frac{m}{t}\right)^t 2^{-t}\left(\frac{2em}{t}\right)^{-t/2} = \frac{2}{t}\left(\frac{m}{8et}\right)^{t/2}$$

where we have used $t \leq m/4$.

Next, note that for any $Q_{\mathbf{v}} \in \mathcal{Q}$, and hence any $Q_{\mathbf{v}} \in \mathcal{Q}'$, we have

$$1 + \chi^2(Q^{\otimes n}\|P^{\otimes n}) = \mathbb{E}_{P^{\otimes n}} \prod_{\mathbf{v}_i=1} \frac{Q_0^{\otimes n}}{P_0^{\otimes n}}(X_{\pi_i}^n) = \left(1 + \chi^2(Q_0^{\otimes n}\|P^{\otimes n})\right)^t.$$

Finally,

$$1 + \chi^2(\langle \mathcal{Q}^{\otimes n}\rangle \| P^{\otimes n}) = \frac{1}{|\mathcal{Q}|^2} \sum_{\mathbf{v},\mathbf{v}' \in \mathcal{V}_t} \mathbb{E}_{P^{\otimes n}}\left[\frac{Q_{\mathbf{v}}^{\otimes n} Q_{\mathbf{v}'}^{\otimes n}}{(P^{\otimes n})^2}(X^n)\right]$$

$$= \frac{1}{\binom{m}{t}^2} \sum_{\mathbf{v},\mathbf{v}' \in \mathcal{V}_t} \prod_{i:\mathbf{v}_i=\mathbf{v}_i'=1} \mathbb{E}_{P_0^{\otimes n}}\left[\frac{(Q_0^{\otimes n})^2}{(P_0^{\otimes n})^2}(X_{\pi_i}^n)\right]$$

$$\leq \frac{1}{\binom{m}{t}^2} \sum_{\mathbf{v},\mathbf{v}' \in \mathcal{V}_t} (1 + a_n)^{|\{i:\mathbf{v}_i=\mathbf{v}_i'=1\}|}$$

$$= \frac{1}{\binom{m}{t}} \sum_{j=0}^{t} \binom{t}{j}\binom{m-t}{t-j}(1 + a_n)^j.$$

Finally, note that the final expression can be written as $\mathbb{E}[(1 + a_n)^{\mathscr{H}}]$ where $\mathscr{H} \sim \mathrm{Hyp}(m, t, t)$. Since hypergeometric random variables are stochastically dominated by the corresponding binomial random variables, we may upper bound the above by the moment generating function of a $\mathrm{Bin}(t, t/m)$ random variable at $(1 + a_n)$ to yield that

$$1 + \chi^2(\langle \mathcal{Q}^{\otimes n}\rangle\|P^{\otimes n}) \leq (1 + (t/m)((1 + a_n) - 1))^t \leq \exp\left(\frac{t^2}{m}a_n\right). \qquad \square$$

### B.1.2 Proof of Theorem 13

*Proof.* It is a classical fact that the $\chi^2$-divergence tensorises as

$$\chi^2(Q_0^{\otimes n}\|P_0^{\otimes n}) = (1 + \chi^2(Q_0\|P_0))^n - 1.$$

The reason for this is that due to independence, $1 + \chi^2(Q_0^{\otimes n}\|P_0^{\otimes n})$ amounts to a product of second moments of relative likelihoods $(Q/P)$ of iid samples.

Thus, since $\chi^2(Q_0\|P_0) \leq \kappa$, we may set $a_n = (1+\kappa)^n - 1$ in Lemma 12. Now, by LeCam's method (Lemma 10), we know that if $R_{\mathrm{GoF}}(t\sigma) < 1/4$ for a given $n$, then using ensemble from Lemma 12, it must hold that

$$\frac{1}{4} \geq 1 - \sqrt{\frac{1}{2}\log\left(1 + \exp\left(\frac{t^2}{m}a_n\right) - 1\right)}$$

$$\iff a_n \geq 2(3/4)^2\frac{m}{t^2}$$

$$\implies (1+\kappa)^n - 1 \geq \frac{m}{t^2}$$

$$\iff n \geq \frac{1}{\log(1+\kappa)}\log\left(1 + \frac{m}{t^2}\right)$$

Thus, the smallest $n$ for which we can test $t\sigma$-changes in $\mathcal{I}_d$ must exceed the above lower bound, giving the stated claim.

The $\mathbb{EOF}$ claim follows similarly. Using the packing change ensemble from Lemma 12, and the lower bound Lemma 11, if the risk is at most $1/4$ for some $n$, then we find that

$$\frac{1}{4} \geq 1 - \frac{1}{|\widetilde{\mathcal{Q}}|} - \sqrt{\frac{(1+a_n)^t - 1}{|\widetilde{\mathcal{Q}}|}}$$

$$\iff (1+a_n)^t \geq 1 + |\widetilde{\mathcal{Q}}|\left(\frac{3}{4} - \frac{1}{|\widetilde{\mathcal{Q}}|}\right)^2$$

$$\iff (1+\kappa)^{nt} \geq 1 + |\widetilde{\mathcal{Q}}|\left(\frac{3}{4} - \frac{1}{|\widetilde{\mathcal{Q}}|}\right)^2$$

$$\iff n \geq \frac{1}{t\log(1+\kappa)}\log\left(|\widetilde{\mathcal{Q}}|\left(\frac{3}{4} - \frac{1}{|\widetilde{\mathcal{Q}}|}\right)^2\right)$$

Now, since $1 \leq t \leq m/16e$, we observe that

$$|\widetilde{\mathcal{Q}}| \geq \frac{2}{t}\left(\frac{m}{8et}\right)^{t/2} \geq \frac{2}{t}\cdot 2^{t/2} \geq 2.5.$$

Thus, $(3/4 - 1/|\widetilde{\mathcal{Q}}|)^2 \geq 1/9$, and the term in the final log above is at least $\log|\widetilde{\mathcal{Q}}|/9$, which in turn is lower bounded by Lemma 12. Thus continuing the above chain of inequalities, we observe that

$$n \geq \frac{1}{\log(1+\kappa)}\cdot\frac{1}{t}\left(\frac{t}{2}\left(\log\left(\frac{m}{8et}\right) - \frac{(2\log(t/2) + 4\log(3))}{t}\right)\right)$$

Finally, since $\log(x)/(x/2) \leq 1/e$, we may $-2(\log(t/2) + 4\log(3))/t \geq -5$. Folding this $-5$ into the log gives $8e^6 \leq 4000$ in the denominator. Finally, again, this tells us that the infimum of the $n$ for which the $\mathbb{EOF}$ risk is small is at least the above lower bound, yielding the claim. $\square$

## B.2 Expanded Lower Bound Theorem Statements and Proofs

We give slightly stronger theorem statements than those in the main text, and give the proofs of the claimed bounds. In all cases the proofs involve the use of Lemma 12 - we describe which widgets are used, and what values of $\sigma, t$ are needed. Then we simply invoke Theorem 13 repeatedly to derive the results.

## B.3 The case $d \leq s \leq cp$

*Proof of Theorem 3.* **High Temperature Bound** This is shown by using the Triangle construction of §D.1.1. This construction amounts to $\sigma = 1$ and $m = \lfloor p/3 \rfloor$. Thus taking $t = s, \mu = \alpha, \lambda = \beta$ and invoking both Proposition 20 and Theorem 13, we find that so long as $p/6 \geq 16es$, the bounds

$$n_{\mathrm{GoF}}(s, \mathcal{I}_d) \geq \frac{1}{C\tanh^2(\alpha)e^{-2\beta}}\log\left(1 + \frac{p}{Cs^2}\right),$$

and similarly

$$n_{\mathrm{EoF}}(s, \mathcal{I}_d) \geq \frac{1}{C \tanh^2(\alpha) e^{-2\beta}} \log\left(\frac{p}{Cs}\right).$$

**Low Temperature Bound** Let $\beta d \geq \log d$. We show this for even $d$ - odd $d$ follows by reducing $d$ by one. We use the Emmentaler clique versus the full clique of §D.2.3 with $\ell = 1$. This corresponds to $\sigma = d/2$ and $m = \lfloor p/d + 1 \rfloor \geq p/2d$. Now take $t = \lceil 2s/d \rceil \leq 4s/d$. Note that the total number of changes is at least $s$ and at most $d/2\lceil 2s/d \rceil \leq 2s$. Notice that $t \leq m$ holds so long as $s \leq p/K$ for some $K \geq 400$. Invoking Proposition 35 in the case of $\mu = \alpha, \lambda = \beta$, and then Theorem 13 with the stated $m, \sigma, t$, gives us the bound

$$n_{\mathrm{GoF}} \geq \frac{1}{Cd^2 \min(1, \mu^2 d^4) e^{-2\beta(d-3)}} \log\left(1 + \frac{1}{C} \frac{(p/2d)}{(4s/d)^2}\right)$$

$$\geq \frac{e^{2\beta(d-3)}}{C'd^2 \min(1, \mu^2 d^4)} \log\left(1 + \frac{1}{C'} \frac{pd}{s^2}\right),$$

where the $(d-3)$ in the exponent arises as $(d-1) - 1 - \ell$, and $d-1$ occurs since we may reduce $d$ by 1 to make it even. Similarly

$$n_{\mathrm{EoF}} \geq \frac{e^{2\beta(d-3)}}{C'd^2 \min(1, \mu^2 d^4)} \log\left(1 + \frac{1}{C'} \frac{p}{s}\right).$$

**Integrating the bounds.** We now note that if $\beta d \leq 3 \log d$, then

$$\frac{e^{2\beta(d-3)}}{d^2 \min(1, \mu^2 d^4)} \leq \frac{e^{2\beta}}{\tanh^2(\alpha)}.$$

Indeed, in this case, $e^{2\beta(d-3)}$ is bounded as $d^6$, and so the left hand size is at most $d^4 \min(1, \alpha^2 d^4)^{-1} \leq \alpha^{-2}$, which is dominated by the right hand side.

On the other hand even if $\beta d \geq 3 \log d$, we may still use the high temperature bound since this is shown unconditionally. Thus, at least so long as we replace the $pd/s^2$ in the low temperature bound by $p/s^2$, we may take the maximum of the expressions in the above bounds to get a concise lower bound - the low temperature term itself only becomes active when $\beta d \leq 3 \log d$, in which case it is known to be true. The claim thus follows. $\square$

## B.4  The case $cp \leq s \leq cpd^{1-\zeta}$

We first state the commensurate $\mathbb{EOF}$ bound -

**Theorem 14.** *In the setting of Theorem 4, we further have that*

*1. If $\alpha d^{1-\zeta} \leq 1/32$ then $n_{\mathrm{EoF}} \geq C \dfrac{1}{d^{2-2\zeta}\alpha^2} \log\left(1 + C\dfrac{pd^{1-\zeta}}{s}\right).$*

*2. If $\beta d \geq 4\log(d-4)$ then $n_{\mathrm{EoF}} \geq C \dfrac{e^{2\beta d(1-d^{-\zeta})}}{d^2 \min(1, \alpha^2 d^4)} \log\left(1 + C\dfrac{pd^{1-\zeta}}{s}\right).$*

*Proofs of Thms. 4 and 14.*
**High Temperature Bounds** Suppose $s = pd^{1-\zeta_0}/K$ for any $\zeta_0 \in (0, 1]$. We invoke the widget of a full $d^{1-\zeta_0}$-clique as $Q_0$ versus an empty graph as $P_0$, i.e. the construction of §D.1.2. This corresponds to taking $\sigma = d^{2-2\zeta_0}/2 + O(d)$, $m \geq pd^{-(1-\zeta_0)}/2$ and $t = \lfloor 2sd^{-(2-2\zeta_0)} \rfloor$, with the total edit made being at most $2s$. Invoking Proposition 21 with $\mu = \alpha$, and then Theorem 13 gives the bounds on noting that

$$\frac{m}{t^2} \geq C \frac{pd^{-(1-\zeta_0)}}{(sd^{-(2-2\zeta_0)})^2} = C \frac{pd^{3-3\zeta_0}}{s^2},$$

$$\frac{m}{t} \geq C \frac{pd^{-(1-\zeta_0)}}{(sd^{-(2-2\zeta_0)})} = C \frac{pd^{1-\zeta_0}}{s^2}$$

and then finally setting $\zeta_0 \geq \zeta$ to derive the claim.

**Low Temperature Bounds** Again fix a $\zeta_0$. We invoke the Emmentaler clique v/s full clique widget of D.2.3, but this time with $\ell = d^{1-\zeta_0}$. This gives $\sigma \approx d^{2-\zeta_0}/2$, $m = \lfloor p/d \rfloor$ and $t = \lceil 2sd^{-2-\zeta_0} \rceil$. The bound now follows similarly to the above section upon invoking Propositions 35 with $\lambda = \beta, \mu = \alpha$ and then Theorem 13 with the stated $m, t, \sigma$. We only track the terms in the log, which are

$$\frac{m}{t^2} \geq C \frac{pd^{-1}}{(sd^{-(2-\zeta_0)})^2} = C \frac{pd^{3-2\zeta_0}}{s^2},$$

$$\frac{m}{t} \geq C \frac{pd^{-1}}{(sd^{-(2-\zeta_0)})} = C \frac{pd^{1-\zeta_0}}{s^2}. \qquad \square$$

## B.5 Proofs in the setting $s \leq d$

The catch in this section is that the Emmentaler clique construction of the proofs above can no longer be employed, since setting even $\ell = 1$ in these induces $\Omega(d)$ changes. We instead turn to the clique with a large hole construction of §D.2.2.

*Proof of Theorem 5.* **High Temperature Bound** This is the same as the high temperature bound of Thm. 3, and that proof may be repeated.

**Low Temperature Bound** Suppose $\beta d \geq 3 \log d$. We use the clique with a large hole construction of §D.2.2 with the choice of $\ell = \lceil \sqrt{2s} \rceil$. This amounts to $s \leq \sigma = s + O(\sqrt{s}) \leq 2s$, and $m = \lfloor p/d \rfloor$. We then simply set $t = 1$ in Theorem 13. Now invoking Proposition 27, we find that

$$n_{\mathrm{GoF}} \geq \frac{1}{C\sqrt{s} \sinh^2(\alpha\sqrt{s}) e^{-2\beta(d-1-2\sqrt{s})}} \log\left(1 + \frac{p}{Cd}\right) \geq \frac{e^{2\beta(d-1-2\sqrt{s})}}{Cd^6 \sinh^2(\alpha\sqrt{s})} \log\left(1 + \frac{p}{Cd}\right)$$

and the same lower bound for $n_{\mathrm{EoF}}$ since in this case $m/t^2 = m/t = 1$ (the $d^6$ is introduced to make the following easy).

**Integrating the bounds** Similarly to the proof of Thm. 3, note that for $\beta d \leq 3 \log d$, $e^{2\beta d} d^{-6} \leq 1$, allowing us to rewrite the low-temperature bound as the max expression in the theorem statement. Giving up space in the logarithm to $p/s^2 \wedge p/d$ then yields the stated claim for $\mathbb{GOF}$. For $\mathbb{EOF}$, we follow the same procedure, but note that since $s \leq d$, $(p/s \wedge p/d) = p/d$. $\qquad \square$

# C Appendix to §4

## C.1 Testing Deletions in Forests, and Changes in Trees

### C.1.1 Proofs of Lower Bounds

*Proof of Lower bounds from Theorem 7.* First note that $n \geq 1$ is necessary, since testing/estimation with no samples is impossible. To derive the second term in the converse for $\mathbb{GOF}$ and the converse for $\mathbb{EOF}$, we plug in the single-edge widget of §D.1.4 with $\mu = \alpha$ into Theorem 13. The widget corresponds to $\nu = 2$ and $\sigma = 1$. Thus, setting $t = s$ and $m = \lfloor p/2 \rfloor \geq p/3$, we obtain both the claimed bounds. $\qquad \square$

### C.1.2 Proof of Upper Bound of Theorem 7, and of Theorem 8

We give the proof for $\alpha > 0$. The proof for $\alpha < 0$ follows identically.

We use $u$ as a short hand for a pair $(i, j)$ with $i < j$, and set $Z_u = X_i X_j$. We exploit two key properties of forest structured graphs

1. For any $u = (i, j)$, if nodes $i$ and $j$ are connected via the graph, then $\mathbb{E}[Z_u] = \prod_{v \in \mathrm{path}(u)} \tanh(\theta_v)$, where for $u = (i, j)$ $\mathrm{path}(u)$ is the unique path connecting $i$ and $j$. If $i$ and $j$ are not connected, then $\mathbb{E}[Z_u] = 0$.

2. For any $u \neq v$, $\mathbb{E}[Z_u Z_v] = \mathbb{E}[Z_u]\mathbb{E}[Z_v]$, that is, the $Z_u$s are pairwise uncorrelated.

The above are standard properties, and are shown by exploiting the fact that conditioning on any node in the forest breaks it into two uncorrelated forests. See, e.g. [BK16] for proofs.

*Proof of Upper Bound in Theorem 7.* Recall the statistic $\mathscr{T} = \sum_{\ell=1}^{n} \sum_{u \in G(P)} Z_u^\ell / n$, where the outer sum is over samples. Suppose $G(P)$ has $k$ edges. Let $\tau := \tanh(\alpha)$. We propose the test

$$\mathscr{T} \underset{\text{Alt}}{\overset{\text{Null}}{\gtrless}} (k - s/2)\tau.$$

Since the sum is over all edges in $p$, and since all edges have the same weight $\alpha$, we note that

$$\mathbb{E}_P[\mathscr{T}] = k\tau.$$

Now consider an alternate $Q_\Delta$ that deletes some $\Delta \geq s$ of these edges. Since a deletion of an edge in the forest disconnects the nodes at the end of the edges (the path connecting two nodes in a forest is unique, if it exists, and we've just removed that unique path by deleting the edge),

$$\mathbb{E}_{Q_\Delta}[\mathscr{T}] = (k - \Delta)\tau.$$

Next, we consider the variance of the statistic. Due to uncorrelation of $Z_u$s, under any forest structured Ising model we have in the case of $n = 1$

$$\mathrm{Var}[\mathscr{T}] = \sum_{u \in G(P)} (1 - (\mathbb{E}[Z_u])^2,$$

where we have used that $Z_u^2 = (\pm 1)^2 = 1$ always. Using the standard behaviour of variances under averaging of independent samples,

$$\mathrm{Var}_{P^{\otimes n}}[\mathscr{T}] = \sum_{u \in G(P)} \frac{1 - \tau^2}{n} = \frac{k(1 - \tau^2)}{n},$$

$$\mathrm{Var}_{Q_\Delta^{\otimes n}}[\mathscr{T}] = \sum_{u \in G(P) \cap G(Q_\Delta)} \frac{1 - \tau^2}{n} + \sum_{u \in G(P) \setminus G(Q_\Delta)} 1/n = \frac{k(1 - \tau^2) + \Delta\tau^2}{n}.$$

Using Tchebycheff's inequality, we then observe that for a given constant $C > 1$, the following hold with probability at least $7/8$ :

$$\text{Under } P^{\otimes n}: \quad \mathscr{T} \geq k\tau - C\sqrt{\frac{k(1 - \tau)^2}{n}},$$

$$\text{Under any } Q_\Delta^{\otimes n}: \quad \mathscr{T} \leq (k - \Delta)\tau + C\sqrt{\frac{k(1 - \tau)^2 + \Delta\tau^2}{n}}.$$

Thus, the test has false alarm and size both at most $1/8$, irrespective of $P$ and $Q_\Delta$, so long as

$$(k - \Delta)\tau + C\sqrt{\frac{k(1 - \tau)^2 + \Delta\tau^2}{n}} < (k - s/2)\tau < k\tau - C\sqrt{\frac{k(1 - \tau)^2}{n}}.$$

Solving out the upper bound on $(k - s/2)\tau$ yields

$$n > 4C^2 \frac{k}{s^2}(\tau^{-2} - 1),$$

while for the lower bound, since $\Delta \geq s$, the same must hold if

$$(k - \Delta)\tau + C\sqrt{\frac{k(1 - \tau)^2 + \Delta\tau^2}{n}} < (k - \Delta/2)\tau,$$

which may be rearranged to

$$n > 4C^2 \left( \frac{1}{\Delta} + \frac{k}{\Delta^2}(\tau^{-2} - 1) \right),$$

which in turn must hold if

$$n > 4C^2 \left(1 + \frac{k}{s^2}(\tau^{-2} - 1)\right),$$

where the final inequality again utilises $\Delta \geq s$.

Thus, forests with $k$ edges can be tested with risk at most $1/4$ as long as we have at least

$$4C^2 \left(1 + \frac{k}{s^2}(\tau^{-2} - 1)\right) + 1 \leq C' \max\left(1, \frac{k}{s^2}(\tau^{-2} - 1)\right)$$

samples, where $C' \leq 8C^2 + 1$ is a constant. Since forests on $p$ nodes have at most $p - 1$ edges, replacing $k$ by $p$ yields an upper bound on the sample complexity of testing deletions in forests.

Finally, since $\tau = \tanh(\alpha)$, we note that $\tau^{-2} - 1 = \sinh^{-2}(\alpha)$, concluding the proof. $\qquad\square$

**Some Observations**

- While the above proof is for uniform edge weights, this can be relaxed with little change. However, the above proof does strongly rely on the edge weights all having the same sign. If this is not the case, then we may encounter edit the same number of positively and negatively weighted edges, and the statistic $\mathscr{T}$ becomes uninformative.

- The statistic $\mathscr{T}$ similarly loses power in the general setting of testing both additions and deletions in forests. This is because while the variance remains controlled as $k(1 - \tau^2)$, the means under the alternates may not move if the only changes being made are additions.

- On the other hand, if we consider testing only of full trees, i.e. $P$ such that $G(P)$ has the full $(p - 1)$ edges, and further the altered $Q$ are also trees, then something interesting emerges - at least in the setting of uniform weights. Since at least $s$ edges were changed from $G(P)$ to $G(Q)$, and one cannot add an edge to $G(P)$ without introducing a cycle, it must be the case that $G(Q)$ effects at least one edge-deletion for every edge it adds, and so it must make at least $\geq s/2$ deletions. In this case, the statistic discussed above *is* powerful. This, of course, was the point of Theorem 8 in the main text, which we are now ready to prove

*Proof of Theorem 8.* Assume that $\alpha > 0$. The proof proceeds similarly for $\alpha < 0$. We use the statistic $\mathscr{T}$ from the proof of the upper bound of Thm. 7 above, and also reuse the notation of $\tau, \Delta$ and $Q_\Delta$ from the above. The claim relies on the above observation that if $\Delta$ edges are changed, then at least $\Delta/2 \geq s/2$ edges must be deleted.

In this case, the mean and the variance of $\mathscr{T}$ under $P$ remain unchanged. On the other hand, under $Q_\Delta$, for any edge $u \in G(P)$ that was deleted in $G(Q_\Delta)$, we must have $|\mathbb{E}_{Q_\Delta}[Z_u]| \leq \tau^2$, since the distance between the end points of these edges is now at least 2. Further, since $G(Q)$ is a tree, the variance of the statistic under $Q_\Delta$ (for $n = 1$) is

$$\begin{aligned}
\mathrm{Var}_{Q_\Delta}[\mathscr{T}] &= \sum_{u \in G(P)} (1 - \mathbb{E}_{Q_\Delta}[Z_u]^2) \\
&\leq (p - 1 - \Delta)(1 - \tau^2) + \Delta \\
&= (p - 1)(1 - \tau^2) + \Delta\tau^2.
\end{aligned}$$

At this point the argument from the earlier proof of Thm. 7 can be used. The test needs to be updated to declaring for the null only when $\mathscr{T} > (p - 1)\tau - s\tau(1 - \tau)/4$. $\qquad\square$

### C.1.3 Tolerant Testing of Forest Deletions, and of Trees

As discussed in the main text, the tolerant testing problem admits as parameters a class of Ising models $\mathcal{J}$, a given Ising model $P \in \mathcal{J}$, a change parameter $s \leq p$, and a tolerance $\varepsilon \in (0, 1)$, with the goal of testing, via sample access, if an unknown model $Q \in \mathcal{J}$ has a network structure that has at most $\varepsilon s$ edges different from $G(P)$, or if it has at least $s$ edges different instead. Concretely, we may define the following risk functions, the latter of which is for the deletion version of tolerant testing:

$$R_{\text{tol}}^{\text{GoF}}(n, s, \varepsilon, \mathcal{J}) = \inf_{\Psi} \sup_{P \in \mathcal{J}} \left\{ \sup_{\widetilde{P} \in A_{\varepsilon s}(P)^c \cap \mathcal{J}} \widetilde{P}^{\otimes n}(\Psi = 1) + \sup_{Q \in A_s(P) \cap \mathcal{J}} Q^{\otimes n}(\Psi = 0) \right\},$$

$$R_{\text{tol}}^{\text{GoF,del}}(n, s, \varepsilon, \mathcal{J}) = \inf_{\Psi} \sup_{P \in \mathcal{J}} \left\{ \sup_{\substack{\widetilde{P} \in A_{\varepsilon s}(P)^c \cap \mathcal{J} \\ G(\widetilde{P}) \subset G(P)}} \widetilde{P}^{\otimes n}(\Psi = 1) + \sup_{\substack{Q \in A_s(P) \cap \mathcal{J} \\ G(Q) \subset G(P)}} Q^{\otimes n}(\Psi = 0) \right\}.$$

Analogously to §2, the sample complexities $n_{\text{GoF}}^{\text{tol}}(s, \varepsilon, \mathcal{J})$ and $n_{\text{GoF,del}}^{\text{tol}}(s, \varepsilon, \mathcal{J})$ are the smallest $n$ required to drive the above risks below $1/4$. Our claim in the main text may be summarised as follows.

**Theorem 15.** *There exists a constant $C$ independent of $(s, p, \alpha, \varepsilon)$ such that*

$$n_{\text{GoF}}^{\text{tol}}(s, \varepsilon, \mathcal{F}(\alpha)) \le C \max \left\{ 1, \frac{1}{\sinh^2(\alpha)} \frac{p}{(1-\varepsilon)^2 s^2}, \frac{1}{(1-\varepsilon)^2 s} \right\}.$$

*Further, if $\varepsilon < {}^{1-\tanh(\alpha)}/_2$, then*

$$n_{\text{GoF}}^{\text{tol}}(s, \varepsilon, \mathcal{T}(\alpha)) \le C \max \left\{ 1, \frac{1}{\sinh^2(\alpha)} \frac{p}{(1-2\varepsilon-\tanh(\alpha))^2 s^2}, \frac{1}{(1-2\varepsilon-\tanh(\alpha))^2 s} \right\}.$$

*Proof.* We repeatedly reuse the notation from the proof of Theorem 7 above.

For the forest deletion setting, suppose $|G(P)| = k$, and let $\widetilde{P}_{\Delta_0}$ be such that it's network structure is a deletion of most $\Delta_0 \le \varepsilon s$ edges from $G(P)$. It follows from the mean and variance calculations before, that, for any $\Delta \ge s$,

$$\mathbb{E}_{\widetilde{P}_{\Delta_0}^{\otimes n}}[\mathscr{T}] = (k - \Delta_0)\tau \ge (k - \varepsilon s)\tau,$$

$$\text{Var}_{\widetilde{P}_{\Delta_0}^{\otimes n}}[\mathscr{T}] = \frac{k(1-\tau^2) + \Delta_0 \tau^2}{n} \le \frac{k(1-\tau^2) + \Delta \tau^2}{n}.$$

Consider the test which rejects the null hypothesis when $\mathscr{T} < (k - \frac{1+\varepsilon}{2}s)\tau$. Comparing the above to a $Q_\Delta$ as in the proof of Theorem 7, and proceeding as in it, we find that the risk is appropriately controlled so long as the following relations hold for every $\Delta_0 \le \varepsilon s$, and $\Delta \ge s$, where $C$ is an absolute constant:

$$n \ge C \frac{k(\tau^{-2} - 1) + \Delta_0}{\left(\frac{1+\varepsilon}{2}s - \Delta_0\right)^2}$$

$$n \ge C \frac{k(\tau^{-2} - 1) + \Delta}{\left(\Delta - \frac{1+\varepsilon}{2}s\right)^2}$$

The right hand sides of the first and second equations above respectively increase and decrease with $\Delta_0$ and $\Delta$. Thus, setting $\Delta_0 = \varepsilon s$ and $\Delta = s$, and taking the maximum possible $k = p$ tells us that the conditions will be met so long as

$$n \ge 4C \frac{(p-1)\sinh^{-2}(\alpha) + s}{(1-\varepsilon)^2 s^2}$$

For the tree case, the same argument follows but with a small change - in the null case, a change of $\Delta_0$ edges can reduce the mean of $\mathscr{T}$ by $\Delta_0 \tau$, but in the alternate, there may exist changes of $\Delta$ edges which only drop the mean of $\mathscr{T}$ by $\Delta/2(\tau - \tau^2)$. Thus, we use the test

$$\mathscr{T} \overset{\text{Null}}{\underset{\text{Alt.}}{\gtrless}} (p-1)\tau - \frac{1+2\varepsilon}{4}s\tau + \frac{s}{4}\tau^2.$$

Continuing similarly, and keeping in mind that the variance of $\mathcal{T}$ after $\Delta$ changes is at most $(p-1)(1-\tau^2) + \Delta\tau^2$, we find that risk of the above test is controlled so long as for every $\Delta_0 \leq \varepsilon s$, and for every $\Delta \geq s$, the following relations hold

$$n \geq \frac{C}{s^2} \frac{p(\tau^{-2}-1) + \Delta_0}{(1 + 2\varepsilon - \tau - 4\Delta_0/s)^2}$$

$$n \geq \frac{C}{s^2} \frac{p(\tau^{-2}-1) + \Delta}{(2\Delta/s(1-\tau) - (1+2\varepsilon-\tau))^2}$$

It is a matter of straightforward computation that if $\varepsilon \leq \frac{1-\tau}{2}$, then the right hand sides of the first and second inequality above respectively increase and decrease with $\Delta_0$ and $\Delta$. Thus, setting $\Delta_0 = \varepsilon s$ and $\Delta = s$, the above holds if

$$n \geq \frac{C}{(1-2\varepsilon-\tau)^2}\left(\frac{p(\tau^{-2}-1)}{s^2} + \frac{1}{s}\right).$$

$\square$

## C.2 Testing Deletions in High-Temperature Ferromagnets

### C.2.1 Proof of achievability

*Proof of the upper bound of Theorem 9.* We follow the strategy laid out in the main text. The proposed test statistic is $\mathcal{T}(\{X^{(i)}\}; P) := \widehat{\mathbb{E}}[\sum_{(i,j)\in G(P)} X_i X_j]$, where the $\{X^{(i)}\}$ are the samples, and $\widehat{E}$ indicates the empirical mean over this data. Concretely, the test is to threshold $\mathcal{T}$ as

$$\mathcal{T} \overset{\text{Null}}{\underset{\text{Alt.}}{\gtrless}} \mathbb{E}_P[\mathcal{T}] - Cs\alpha,$$

where $C$ the constant left implicit in Lemma 16.

The analysis relies on two facts:

**Lemma 16.** *Let $P, Q \in \mathcal{H}_d^\eta(\alpha)$, and $G(Q) \subset G(P)$, with $|G(P) \triangle G(Q)| \geq s$. For every $\eta < 1$, there exists a constant $C > 0$ that does not depend on $(p, s, \alpha)$ such that*

$$\mathbb{E}_P[\mathcal{T}] - \mathbb{E}_Q[\mathcal{T}] \geq 2Cs\alpha.$$

**Lemma 17.** *For any $P, Q \in \mathcal{H}_d^\eta(\alpha)$, which may be equal,*

$$\text{Var}_Q\left[\sum_{(i,j)\in G(P)} X_i X_j\right] \leq C_\eta pd,$$

*where $C_\eta$ may depend on $\eta$, but not otherwise on $(p, d, s, \alpha)$.*

Applying the variance contraction over $n$ independent samples, we find via a use of Tchebycheff's inequality that the following event have probability at least $1/8$ for the respective hypotheses:

$$\text{Null:} \quad \mathcal{T} \geq \mathbb{E}_P[\mathcal{T}] - C_\eta\sqrt{\frac{8pd}{n}},$$

$$\text{Alt:} \quad \mathcal{T} \leq \mathbb{E}_P[\mathcal{T}] - Cs\alpha + C_\eta\sqrt{\frac{8pd}{n}}.$$

Thus, taking $n$ so large that $Cs\alpha > C_\eta\sqrt{\frac{8pd}{n}}$, the false alarm and missed detection probabilities are both controlled below $1/8$, yielding the claimed result. $\square$

It of course remains to argue the above lemmata. These are both essentially utilisations of existing results.

*Proof of Lemma 16.* We use the fact that in ferromagnetic models, the correlations between any pair of nodes increases as the weights increase (or contrapositively, if weights are deleted, then correlations must decrease). This is classically shown via (a special case of) Griffith's inequality [Gri69], which claims that for any $u, v, i, j$, in a ferromagnetic Ising model, $\mathbb{E}[X_u X_v X_i X_j] \geq \mathbb{E}[X_u X_v]\mathbb{E}[X_i X_j]$. This is relevant here due to the fact that

$$
\begin{aligned}
\partial_{\theta_{ij}} \mathbb{E}_{P_\theta}[X_u X_v] &= \partial_{\theta_{ij}} \frac{1}{Z_\theta} \sum_x x_u x_v \exp\left(\sum_{s<t} \theta_{st} X_s X_t\right) \\
&\stackrel{a}{=} \frac{1}{Z_\theta} \sum_x x_u x_v x_i x_j \exp\left(\sum_{s<t} \theta_{st} X_s X_t\right) \\
&\quad - \frac{1}{Z_\theta^2}\left(\sum_x x_u x_v \exp\left(\sum_{s<t}\theta_{st}X_s X_t\right)\right)\left(\sum_x x_u x_v \exp\left(\sum_{s<t}\theta_{st}X_s X_t\right)\right) \\
&= \mathbb{E}[X_u X_v X_i X_j] - \mathbb{E}[X_u X_v]\mathbb{E}[X_i X_j] \geq 0.
\end{aligned}
$$

Above, equality $(a)$ is a consequence of the quotient rule, and the fact that $Z_\theta = \sum_x \exp\left(\sum_{s<t} \theta_{st} x_s x_t\right)$.

Next, we utilise the following structural lemma, due to Santhanam and Wainwright. While we cite it as their Lemma 6 below, but more accurately this arises via a correction of a subsidiary part of the proof of the same lemma. In particular, we are utilising a corrected version of the unlabelled inequality on Page 4131 that follows the inequality (51), with further specialisation to the high-temperature deletion with a uniform edge weight context.

**Lemma 18.** (A variation of Lemma 6 of [SW12]) *Let $P \in \mathcal{H}_d^\eta(\alpha)$, and $Q$ be obtained by removing the edge $(a, b)$ from $P$. Then*

$$
\mathbb{E}_P[X_a X_b] - \mathbb{E}_Q[X_a X_b] \geq \frac{\alpha}{400}.
$$

With this in hand, we develop our result by arguing over each deleted edge in a sequence. For a given $P$ and $Q$, such that $Q$ occurs by deleting $\Delta \geq s$ edges from $P$, take a chain of laws $P = Q_0, Q_1, Q_2, \ldots, Q_\Delta = Q$, where each $Q_{t+1}$ is obtained by deleting one edge from $Q_t$. Let $(i_{t+1}, j_{t+1})$ be the edge deleted in going from $Q_t$ to $Q_{t+1}$ Since each model is ferromagnetic, and each $Q_{t+1}$ deletes an edge from $Q_t$, we find that

$$
\mathbb{E}_{Q_t}\left[\sum_{(i,j)\in G(P)} X_i X_j\right] - \mathbb{E}_{Q_{t+1}}\left[\sum_{(i,j)\in G(P)} X_i X_j\right] \geq \mathbb{E}_{Q_t}\left[X_{i_{t+1}}X_{j_{t+1}}\right] - \mathbb{E}_{Q_{t+1}}\left[X_{i_{t+1}}X_{j_{t+1}}\right]
$$

$$
\geq \frac{\alpha}{400}.
$$

Summing up the left hand side over $t = 0$ to $\Delta - 1$ leads to a telescoping sum, while $s$ copies of the right hand side get added, directly leading to our conclusion

$$
\begin{aligned}
\mathbb{E}_P\left[\sum_{(i,j)\in G(P)} X_i X_j\right] - \mathbb{E}_Q\left[\sum_{(i,j)\in G(P)} X_i X_j\right] & \\
&= \mathbb{E}_{Q_0}\left[\sum_{(i,j)\in G(P)} X_i X_j\right] - \mathbb{E}_{Q_\Delta}\left[\sum_{(i,j)\in G(P)} X_i X_j\right] \\
&= \sum_{t=0}^{\Delta-1} \mathbb{E}_{Q_t}\left[\sum_{(i,j)\in G(P)} X_i X_j\right] - \mathbb{E}_{Q_{t+1}}\left[\sum_{(i,j)\in G(P)} X_i X_j\right] \\
&\geq \sum_{t=0}^{\Delta-1} \frac{\alpha}{400} = \Delta\frac{\alpha}{400} \geq s\frac{\alpha|}{400}. \qquad \square
\end{aligned}
$$

To complete the proof, we prove the key lemma used in the above argument.

*Proof of Lemma 18.* We note that this proof assumes familiarity with the proof of Lemma 6 of [SW12]. The main reason is that the proof really consists of fixing an equation in the proof of this result, and then utilising the ferromagnetic properties a little. As a result, there is no neat way to make this proof self contained (reproducing the proof of the aforementioned lemma is out of the question, since this is a long and technical argument in the original paper). With this warning out of the way, let us embark.

Let $\partial a$ and $\partial b$ be the neighbours of, respectively, $a$ and $b$ in $G(P)$ (which, since $G(Q)$ only deletes $(a, b)$ from $G(P)$, contain all the neighbours of $a$ and $b$ in $G(Q)$ as well).

Before proceeding, we must first point out a (small) error in the proof of Lemma 6 in [SW12]. The clearest way to see this error is to note the inequality following equation (51) in the text, which claims that if $(a, b) \in G(P) \triangle G(Q)$, then some quantity ($J$ in the paper) known to be positive is upper bounded by

$$J \leq \sum_{u \in \partial a \setminus \{b\}} (\{\mathbb{E}_P - \mathbb{E}_Q\}[X_u X_a])(\theta_{ua}^P - \theta_{ua}^Q) + \sum_{v \in \partial b \setminus \{a\}} (\{\mathbb{E}_P - \mathbb{E}_Q\}[X_v X_b])(\theta_{vb}^P - \theta_{vb}^Q).$$

Note that we have specialised the above to the case where $G(Q) \subset G(P)$. Now, observe than when the only change made is in the edge $(a, b)$, then the above upper bound is 0. Indeed, $\theta_{ua}^P = \theta_{ua}^Q$ for every $u \in \partial a \setminus \{b\}$, since none of these edges have been altered, making the first sum 0, and similarly the second, contradicting the claim that the sum is bigger than $J$ (which is positive). The error actually lies a few lines up, in the decomposition for the term $\Delta(\theta, \theta')$, which along with the claimed terms, should also include the term $(\{\mathbb{E}_P - \mathbb{E}_Q\}[X_a X_b])(\theta_{ab}^P - \theta_{ab}^Q)$, which is missing from the text of [SW12]. This term is present since the $P_{\theta[x_C]}$ and $P_{\theta'[x_C]}$ are, of course, laws on $X_a$ and $X_b$, and thus have $\theta_{ab}^P x_a x_b$ and $\theta_{ab}^Q x_a x_b$ in the Ising potentials.[3] Putting this term back in, the correct equation is that

$$\kappa \leq (\{\mathbb{E}_P - \mathbb{E}_Q\}[X_a X_b])(\theta_{ab}^P - \theta_{ab}^Q) + \sum_{u \in \partial a \setminus \{b\}} (\{\mathbb{E}_P - \mathbb{E}_Q\}[X_u X_a])(\theta_{ua}^P - \theta_{ua}^Q)$$
$$+ \sum_{v \in \partial b \setminus \{a\}} (\{\mathbb{E}_P - \mathbb{E}_Q\}[X_v X_b])(\theta_{vb}^P - \theta_{vb}^Q),$$

where $\kappa$ is the lower bound on $J$, that is (specialised to our case of uniform weights),

$$\kappa = \frac{\sinh^2(\alpha/4)}{1 + 3\exp(\alpha d)}.$$

We note that the conclusion of Lemma 6 of [SW12] is not affected by the above error[4].

With this out of the way, we may now argue our point. In our case, we know that since only the edge $(a, b)$ has been altered, the second and third terms in the updated sum are 0. Further, we know that $\theta_{ab}^P = \alpha \geq 0$, and $\theta_{ab}^Q = 0$. Thus, we conclude that

$$\mathbb{E}_P[X_a X_b] - \mathbb{E}_Q[X_a X_b] \geq \frac{\kappa}{\alpha} \geq \frac{\sinh^2 \alpha/4}{\alpha(1 + 3\exp(2\alpha d))}.$$

Finally, we use our high temperature condition. Firstly, note that $\alpha d \leq \eta < 1$, and thus $(1 + 3\exp(2\alpha d)) \leq 1 + 3e^2 \leq 24$. Next, since $\sinh(x) \geq x$, $\sinh^2(\alpha/4) \geq \alpha^2/16$. Putting these together, we find that

$$\mathbb{E}_P[X_a X_b] - \mathbb{E}_Q[X_a X_b] \geq \frac{\alpha^2/16}{\alpha \cdot 24} = \frac{\alpha}{384} \geq \frac{\alpha}{400} \qquad \square$$

*Proof of Lemma 17.* We directly utilise the concentration result [Ada+19, Ex. 2.5], which shows that for bilinear forms $f(X) = \langle A, XX^\top \rangle$, where the inner product is the Frobenius dot product, and for a high temperature Ising model $P$, there exists a $C_\eta$ depending only on $\eta$ such that[5]

$$P(|f - \mathbb{E}[f]| \geq t) \leq 2 \exp\left(-\frac{t}{C_\eta \|A\|_F}\right).$$

Via the standard integral representation $\mathbb{E}[(f - \mathbb{E}[f])^2] = \int_0^\infty P(|f - \mathbb{E}[f]|^2 \geq r)\mathrm{d}r$ and the above upper bound, we directly obtain that the variance of any $f$ such as the above is bounded by $3\|A\|_F^2 C_\eta^2$.

Now, out statistic is a bilinear function of the above form. Indeed,

$$\sum_{(i,j) \in G(P)} X_i X_j = \langle G(P)/2, XX^\top \rangle,$$

where we treat $G(P)$ as it's adjacency matrix, and thus we immediately obtain that the variance is bounded by $1.5 C_\eta^2 \|G(P)\|_F^2$. Notice that $\|G(P)\|_F^2$ is merely twice the number of edges in $G(P)$, and since this has degree at most $d$, this number is at most $2pd$. The claim follows. $\square$

### C.2.2 Proof of Lower Bounds

The lower bounds are argued using Thm. 13, with the widget(s) that consist of comparing a full clique to an empty graph, which of course satisfy the constraint that the alternate models are derived by deleting edges from the null graph. Concretely, we use the bound of Proposition 22, to show the following result

**Proposition 19.** *Suppose $s \leq pd/K$ for large enough $K$ and $\alpha d \leq \eta \leq 1/32$. Then there exists a $C$ independent of all parameters such that*

$$n_{\mathrm{GoF,del}}(s, \mathcal{H}_d^\eta(\alpha)) \geq \max_{s/Kp \leq k \leq d} \frac{1}{Ck^2\alpha^2} \log\left(1 + \frac{pk^3}{Cs^2}\right),$$

$$n_{\mathrm{GoF,del}}(s, \mathcal{H}_d^\eta(\alpha)) \geq \max_{s/Kp \leq k \leq d} \frac{1}{Ck^2\alpha^2} \log\left(1 + \frac{pk}{Cs}\right),$$

*where the maximisation is over integers $k \geq 2$ in the stated ranges. In particular, the bounds in the main text correspond to taking $k = d$.*

*Proof.* The proof relies on the fact that if $\alpha d \leq 1/32$, then $\alpha k \leq 1/32$ for any $k \leq d$ as well, which allows us to utilise Prop. 22 for each $k$. For each valid choice of $k$, we take $P_0$ to be the Ising model on the complete graph on $k$ nodes with uniform edge weight $\alpha$, and $Q_0$ to be the Ising model on the empty graph on $k$ nodes. The relevant quantities are $\sigma = \binom{k}{2}$, $m = \lfloor p/k \rfloor$, and $t = \lceil s/\binom{k}{2} \rceil$, with the total number of changes lying between $s$ and $2s$. Repeated use of Thm.13 concludes the argument. $\square$

### C.3 Simulation Details

Details about the generation of Figure 3 are as follows:

- **Sampling from Ising Models** Samples from Ising models were generated by running Glauber dynamics for 1600 steps. This number is chosen to be four times the 'autocorrelation time', which is the time at which the autocorrelation of the test statistic $\langle XX', G \rangle/2$ drops to near 0, and serves as a proxy for the mixing time of the dynamics (at least for the relevant statistics). Note that raw samples were outputted from the dynamics (i.e., we did not take ergodic averages).

- **Constructing $P$s and $Q$s** Throughout, $P$ was the Ising model on a complete binary tree on 127 nodes. For each value of $s$ and each experiment, $s$ random edges from this tree were deleted.

Figure 4: Reconstruction Error of the Chow-Liu Tree for the Ising model on a complete Binary Tree with $p = 127, \alpha = 0.1$.

- **Experiment Structure** For each value of $s \in \{3, 6, \ldots, 60\}$ and $n \in \{20, 40, \ldots, 480\}$, we carried out a simulation of the GoF testing risk of our statistic for $s$ deletions using $n$ samples. We refer to each of these as an experiment. Each experiment was carried out by running 100 independent tests (on independent data), which each consisted of two parts - first we generated samples from $P$, and declared a false alarm if $\mathscr{T}$ fell below $(p - 1 - s/2) \tanh(\alpha)$ for this. Next, we generated a $Q$ by deleting $s$ edges, and then generated samples from $Q$, and finally declared a missed detection if $\mathscr{T}$ was above the same threshold. Risks were computed by adding up the total number of false alarm and missed detection events in these 100 runs, and dividing them by 100.

- **Structure of Figure 3** Each box in the figure corresponds to a simulation for $s$ changes and $n$ nodes, where $(s, n)$ are the coordinates of the upper right corner of the box. The boxes are coloured according to their empirical risk - if this was greater than $0.35$, then the box was coloured black; if smaller than $0.15$, then coloured white, while if it was between these values, the box was coloured orange.

Additionally, we note that structure learning performs very poorly for this setup. This is best illustrated by the Figure 4, which shows the number of edge-errors (i.e. $|G(P) \triangle \hat{G}|$) versus the sample size when the Chow-Liu algorithm was run on data generated by the null model (i.e., the full binary tree). The Chow-Liu algorithm was run by computing the covariance matrix, and computing the weighted maximum spanning tree for it via the library methods in MATLAB. The number of errors is again averaged over 100 trials. This demonstrates that the naïve scheme of recovering the graph and testing against it is infeasible for $s \leq 60$ if $n \leq 1500$, empirically demonstrating the separation between structure learning and testing.

# D    Widgets

As discussed in the previous section, we will utilise Lemma 6, in order to do which we need to provide specific instances of $(P_0, Q_0)$ that are close in $\chi^2$-divergence. We will abuse terminology and call this pair an ensemble. This section lists a few such pairs of graphical models, along with the $\chi^2$-divergence control we offer for the same, proofs for which are left to §F. Throughout, we will use $\lambda$ and $\mu$ as weights of edges, with $\lambda \geq |\mu| > 0$. I the proofs of the theorems, we will generally set $\lambda = \beta$ and $\mu = \alpha$, but retaining these labels aids in the proofs of $\chi^2$-divergence control offered for these widgets.

## D.1    High-Temperature Obstructions

The following graphs are used to construct obstructions in high temperature regimes. The first is the triangle graph, as described in §3.1. The second is a full clique in high temperatures. The latter section is derived from the bounds of [CNL18].

### D.1.1 The Triangle

We start simple. Let $P_{\text{Triangle}}$ be the Ising model on 3 nodes with edges $(1,2)$ and $(2,3)$, each with weight $\lambda$, and $Q_{\text{Triangle}}$ be the same with the edge $(1,3)$ of weight $\mu$ appended (see Figure 5). The bound below follows from an explicit calculation, which is tractable in this small case.

Figure 5: Ensemble used for Proposition 20

**Proposition 20.** *For $\lambda \geq |\mu| > 0$,*

$$\chi^2(Q_{\text{Triangle}}\|P_{\text{Triangle}}) \leq 8e^{-2\lambda}\tanh^2 \mu.$$

### D.1.2 Full Clique versus Empty Graph

[CNL18] shows the remarkable fact that high-temperature cliques are difficult to separate from the empty graph. We present this result below.

**Proposition 21.** *Let $P$ be the Ising model on the empty graph with $k$ nodes, and let $Q$ be the Ising model on the $k$-clique, with uniform edge weights $\mu$. If $32\mu k \leq 1$, then*

$$\chi^2(Q\|P) \leq 3k^2\mu^2.$$

In the notation of [CNL18], this is the bound at the bottom of page 22, instantiated with $G = G'$ and the $\mathcal{R}, \mathcal{B}, \Gamma$ values as determined in the proof of Example 2.7.

We will also utilise the following reversed $\chi^2$-divergence bound. This is not formally shown in [CNL18], and thus, we include a proof of the same, using the techniques of the cited paper, in §F.2.5.

**Proposition 22.** *Let $P$ be the Ising model on a clique on $m$ nodes with uniform edge weights $\mu$, and let $Q$ be the Ising model on the empty graph on $m$ nodes. If $32\mu m \leq 1$, then*

$$\chi^2(Q\|P) \leq 8(\mu m)^2.$$

### D.1.3 Fan Graph

This widget is not required for the main text, although it may serve as a more involved construction to show the bounds of Thms. 3 and 5. Its main use is in Appendix E.2, where it is used to show an obstruction to testing of maximum degree in a graph.

Generalising the triangle of the previous section, we may hang many triangles from a single vertex, getting a graph that resembles an axial fan with many blades. Using such a graph, we may demonstrate high-temperature obstructions to determining the maximum degree of a graph.

Concretely, for a natural $B$ we define a fan with $B$ blades to be the graph on $2B + 1$ nodes where, nodes $[1 : 2B]$ are each connected to the central node $2B + 1$, and further, for $i \in [1 : B]$, nodes $2i$ and $2i - 1$ are connected. We call the edges incident on the central node $(B + 1)$ axial, and the remaining edges peripheral.

Treating $\ell$ as a parameter, the Ising models $P_{\ell,\text{Fan}}$ and $Q_{\ell,\text{Fan}}$ are determined as followed:

- $Q_{\ell,\text{Fan}}$ places a weight $\lambda$ on each peripheral edge, and a weight of $\mu$ on each axial edge.
- $P_{\ell,\text{Fan}}$ 'breaks in half' $\ell$ of the blades in the graph - concretely, for $i \in [1 : \ell]$, we delete the edges $\{2i - 1, 2B + 1\}$.

Viewing $P$ as the null graph, note that in $Q$ we have added an excess of $\ell$ edges, and increased the degree of the central node from $2B - \ell$ to $2B$. The fan graph serves as a high-temperature obstruction to determining the maximum degree of the graph underlying an Ising model via the following claim.

Figure 6: The Fan graphs for $P_{\ell,\text{Fan}}$ (left) and $Q_{\ell,\text{Fan}}$ (right) in the setting $B = 4, \ell = 2$.

**Proposition 23.** *For $\ell \leq B$, if $\lambda\mu \geq 0$, then*

$$\chi^2(Q_{\ell,\text{Fan}} \| P_{\ell,\text{Fan}}) \leq \left(1 + 16e^{-2\lambda}\tanh^2\mu\right)^{\ell} - 1.$$

#### D.1.4 Single Edge

This construction is possibly the simplest, and is used to show the lower bound in Thm. 7. We consider the two possible Ising models on two nodes - $P$ is the one with an edge, of weight $\mu$, while $Q$ has no edges. The characterisation is trivial, and we omit the proof.

**Proposition 24.** $\chi^2(Q\|P) = \sinh^2(\mu)$.

### D.2 Low-Temperature Obstructions via Clique-Based Graphs

The computations in this and the subsequent cases are rather more complicated that in the previous case, and will intimately rely on a 'low temprature' assumption. The basic unit is that of a clique on some $d + 1 \gg 1$ nodes, in the setting of temperature $\lambda d \geq \log d$.

The intuition behind these is rather simple - Ising models on cliques tend to 'freeze' in low temprature regimes, i.e. the distribution concentrates to the states $\pm(1, 1, \ldots, 1)$ with probability $1 - \exp\left(-\Omega(\beta d)\right)$ for $\beta d \gg 1$. This effect is fairly robust, and dropping or adding even a large number of edges does not alter it significantly. Thus, one has to collect an exponential in $\beta d$ number of samples merely to obtain some diversity in the samples, which will be necessary to distinguish any of these variations of a clique from the full thing.

While generic arguments can be offered for each of the settings below on the basis of the above intuition, these tend to be lossy in how they handle the effect of very low edge weights. To counteract this, we individually analyse each setting, and while the arguments have structural similarities, the particulars vary.

It is worth noting that our bounds rely on below diverge from the classical literature in the low temperature condition we impose. We generally demand conditions like $\beta d \geq \log d$, while most other lower bounds demand that $\beta d \geq 1$. This extra room allows us to tighten the exponents in the sample complexity bounds as opposed to previous work, but has the obvious disadvantage of reduced applicability. We note, however, that in most settings, this only yields a lost factor of $d$ in the resulting bounds, and frequently not even that. Functionally, thus, there is little to no loss in the use of this stronger low-temperature condition.[6] A similar notion of low temperature has appeared in e.g. [Bez+19].

#### D.2.1 Clique with a deleted edge

This calculation is the simplest demonstration of our bounding technique, and all following settings are analysed in a similar way. While it is superseded by the section immediately following it, the bound is thus important for the reasons of comprehension if nothing else.

We consider graphs on $d + 1$ nodes, and let $P_{\text{Clique}}$ be the Ising model on the complete graph on $d + 1$ nodes, with edge $(1, 2)$ of weight $\mu$, and every other edge of weight $\lambda$. $Q_{\text{Clique}}$ is formed by

deleting the edge $(1,2)$ in $P_{\text{Clique}}$ Note that such underlying constructions feature in nearly every

Figure 7: The clique with uniform weight $\lambda$ barring one edge, and the same edge deleted. Here $d = 4$.

lower bound on structural inference on degree bounded Ising models.

With the exposition out of the way, we state the bound below.

**Proposition 25.** *Suppose $\lambda d > \log d$. Then*

$$\chi^2(Q_{\text{Clique}}\|P_{\text{Clique}}) \leq 16e^{-2\lambda(d-1)}\sinh^2\mu.$$

### D.2.2 The clique with a large hole

To allow for a greater number of changes, we modify the previous construction by removing a large subclique from the $K_{d+1}$ used above, instead of just one edge. More formally, for some $\ell < d/8$, let $K_\ell$ be the complete graph on nodes $[1 : \ell]$. We set $P_{\ell,\text{Clique}}$ to the the Ising model on $K_{d+1}$ such that the edges in $K_\ell$ have weight $\mu$, and all other edges have weight $\lambda$, while $Q_{\ell,\text{Clique}}$ instead deletes the edges in $K_\ell$. Note that as a conseuquence, we have effected a deletion of $\sim \ell^2/2$ edges from the original model.

**Proposition 26.** *If $\ell + 1 \leq d/8$, $\lambda \geq |\mu|$ and $\lambda d > 3\log d$, then*

$$\chi^2(Q_{\ell,\text{Clique}}\|P_{\ell,\text{Clique}}) \leq 32\ell e^{-2\beta(d+1-\ell)}\sinh^2(\mu(\ell-1)).$$

Note that the bound of the previous subsection (up to some factors) can be recovered by setting $\ell = 2$ in the above.

Control on the $\chi^2$-divergence with $P$ and $Q$ exchanged is also useful.

**Proposition 27.** *If $\ell + 1 \leq d/12$, $\lambda \geq |\mu|$ and $\lambda d > 3\log d$, then*

$$\chi^2(P_{\ell,\text{Clique}}\|Q_{\ell,\text{Clique}}) \leq 64\ell e^{-2\beta(d+1-\ell)}\sinh^2(2\mu(\ell-1)).$$

### D.2.3 Emmentaler Clique

As a development of the Clique with a large hole, we may in fact put in many large holes, leading to a pockmarked clique reminiscent of a Swiss cheese. Concretely, let $\ell$ be a number such that $B := d/(\ell+1)$ is an integer. We define a graph on $d$ nodes in the following way: Divide the nodes into $B$ groups of equal size, $V_1, \ldots, V_B$. Form the complete graph on $d$ nodes, and then delete the $\ell+1$-sublique on $V_i$ for each $i$. Note that equivalently, the graph above is the complete symmetric $B$-partite graph on $d$ nodes. The graph effects a deletion of $\sim d\ell/2$ edges from a clique.

The key property of the Emmentaler is that it still freezes at a exponential rate, and it has sufficient 'space' in it to accommodate significantly more edges. In particular, the graph is regular and the degrees of each node are uniformly $d - \ell - 1$. We use this in two ways:

**Emmentaler with one extra node** We show that determining the degree of a node connected to many of the nodes of an Emmentaler is hard. Concretely, we construct the following two graphs on $d + 1$ nodes:

Construct an Emmentaler Clique on the first $d$ nodes. Next, connect the node $d + 1$ to each node in $\bigcup_{i=1}^{B-1} V_i$. Notice that node $d + 1$ is not connected to one of the parts of the Emmentaler. We choose

Figure 8: Two views of the Emmentaler cliques. The left represents the base clique as the large grey circle, while the uncoloured circles within represent the groups $V_i$ with no edges within (this should be viewed as $\ell \gg 1, B = 10$). This view is inspiration for the name. On the right, we represent the Emmentaler as the graph $K_{\ell+1,\ell+1,\ldots,\ell+1}$ - here $d = 8$ and $\ell = 1$ is shown.

$P_\ell$ to be the Ising model with uniform weight $\lambda$ on the this graph. For $Q_\ell$, we additionally add edges between node $d + 1$ and each node in $V_B$ with weight $\mu$. The following result holds.

**Proposition 28.** *If* $2 \le \ell + 1 \le d/4$ *and* $\lambda(d - 4) \ge 3 \log d$, *and* $|\mu| \le \lambda$, *then*

$$\chi^2(Q_\ell \| P_\ell) \le 32 d e^{-2\lambda(d-1-\ell)}.$$

Notice that the above proposition does not show a $\mu$ dependence. This is due to inefficiencies in our proof technique. We strongly conjecture that a bound of the form $(1 + Cd \tanh^2(\mu(\ell + 1))e^{-2\lambda(d-\ell-1)})^n$ holds.

**Emmentaler v/s Full Clique**   We show that it is difficult to distinguish between an Emmentaler and a full clique. Concretely, we let $P_\ell$ be an Emmentaler as above, and in $Q_\ell$, we add back the deleted subcliques to each $V_i$, but with weight $\mu$.

**Proposition 29.** *If* $\ell + 1 \le d/4$ *and* $\lambda(d - 4) \ge 3 \log d$, *then*

$$\chi^2(Q_\ell \| P_\ell) \le d^2 \min(1, \mu^2 d^4) e^{-2\lambda(d-1-\ell)}.$$

# E   Miscellaneous

## E.1   Using statistical formulations to test structural changes

The main text makes the case that statistical formulations of $\mathbb{GOF}$ do not give us the whole story when one is interested in structural changes. Concretely, though, this only directly affects the lower bounds. On the other hand, when we restrict alternate hypotheses in the $\mathbb{GOF}$ problem to make a lot of changes, then one may expect that tests under statistical formulations are powerful.

Intuitively, this expectation is rendered plausible by the fact that the notion of being close to a given model is similar under the statistical and the structural formulations - equality under one is also equality under the second, at least in the setting of unique network structures, and mere continuity suggests that, at least locally, setting some value of $s(P, \varepsilon)$ or $\varepsilon(s, P)$ should allow one to translate tests from the statistical to the structural notions of changes and vice versa.[7] However, this strategy does not work too well, at least with our current understanding of Ising models. More concretely - utilising statistical tests for structural testing in a sample efficient way requires a *local* understanding of the distortion of the edge-Hamming distance of the graph under the map $(\theta, \theta') \mapsto \text{SKL}(\theta \| \theta')$,

which is not available as of now. Global constraints on the same are available, and are unhappily both rather pessimistic, and essentially tight. This means that using the methods developed for testing for statistical divergences in the setting of structural identity testing is problematic.

Some details - the best available results that translate edge-differences to symmetrised KL divergence is via Lemma 4 of [SW12]. The Bhattacharya coefficient of two distributions is $\mathrm{BC}(P, Q) := \sum_x \sqrt{P(x)Q(x)}$. The cited lemma argues that under $s$ changes,

$$\mathrm{BC} \le \exp\left(-Cs \sinh^2(\alpha) e^{-2\beta d}/d\right).$$

Let $-\varphi$ denote the exponent in the above, for conciseness. Since $-2 \log \mathrm{BC} \le \mathrm{KL}$, this induces $D_{\mathrm{SKL}} \gtrsim \varphi$, and similarly, since $1 - \mathrm{BC} \le \mathrm{TV}$, this tells us also that $\mathrm{TV} \ge 1 - \exp(-\varphi)$. Since $1 - e^{-z} \le z$, this means that the best lower bound we can possibly derive this way is $\mathrm{TV} \ge \varphi$.

Now, the best known upper bounds for statistical testing under SKL is $(\beta pd/\varepsilon)^2$ up to log factors [DDK16], and under TV for ferromagnets this may be improved to $(pd/\varepsilon)^2$ [Bez+19]. Plugging in the values of $\varepsilon$ implicit in the above, the first of these then requires about

$$\left(\frac{\beta pd}{\varphi}\right)^2 \sim \frac{e^{4\beta d}}{\alpha^4}\left(\frac{\beta pd^2}{s}\right)^2,$$

which is worse than the testing by first recovering the underlying network. Similarly, under TV, a similar number is required, but without an extra $\beta$-factor, which has little effect in light of terms like $e^{\beta d}$ showing up. So, naïvely using this structural characterisation does not give promising results.

Further, unfortunately, the characterisation of $\mathrm{BC}$, and indeed of KL and TV divergences offered through this is essentially tight. This essentially follows from our results providing control on the $\chi^2$-divergences in various construction, and the control this imposes on $\mathrm{KL}, \mathrm{TV}$ via the monotonicity of Rényi divergences and Pinsker's inequality. It may be the case that in some special cases, tight bounds for structural testing may be derived via the statistical testing approach above. We have not explored this possibility in detail.

### E.2  Lower Bounds on Property Testing

In passing, we mention that our constructions improve upon lower bounds for some of the property tests studied in [NL19]. For instance, the triangle construction provides an obstruction to cycle testing that does not require explicit control on $\alpha$ as in [NL19]. Similarly, the Clique with a hole, and the Emmentaler clique with an extra node constructions may serve as obstructions to testing the size of the largest clique, and to testing the value of the maximum degree of the network structures in low temperatures. In high temperatures, the Fan graph construction shows that testing maximum degree is hard. In each case this either improves upon the lower bounds of [NL19] by either improving the exponent from $\beta d/4$ to $2\beta d(1 - o_d(1))$, or by removing an explicit high-temperature condition that is enforced in the lower bound.

## F  Proofs of Widget Bounds

**An Observation** For Ising models $P, Q$,

$$1 + \chi^2(Q\|P) = \sum_x \frac{Q(x)^2}{P(x)} = \sum_x \frac{Z_P}{Z_Q^2} \exp\left(x^T 2\theta_Q x - x^T \theta_P x\right) = \frac{Z_P Z_{2Q-P}}{Z_Q^2},$$

where $Z_{2Q-P} := \sum_x \exp\left(x^T(2\theta_Q - \theta_P)x\right)$ is yet another partition function. We will repeatedly use this form of the $\chi^2$-divergence, without further comment, in the following.

### F.1  Star-Based Widgets

#### F.1.1  Triangle

*Proof of Proposition 20.* Let $P = P_{\mathrm{Triangle}}, Q = Q_{\mathrm{Triangle}}$. Note that

$$P(x) = \frac{1}{Z_P} e^{\lambda x_2(x_1+x_3)}$$

$$Q(x) = \frac{1}{Z_Q(\mu)} e^{\lambda x_2(x_1+x_3)} e^{\mu x_1 x_3}$$

Where the partition functions may simply be computed to obtain the expressions below:

$$Z_P = 2^3 \cosh^2 \lambda = 4(\cosh 2\lambda + 1)$$
$$Z_Q(\mu) = 4(e^\mu \cosh 2\lambda + e^{-\mu}).$$

Further, we have that

$$W := \mathbb{E}_P[(Q/P)^2] = \left(\frac{Z_P}{Z_Q(\mu)}\right)^2 \cdot \frac{1}{Z_P} \cdot \sum e^{\lambda x_2(x_1+x_3)} e^{2\mu x_1 x_3} = \frac{Z_P Z_Q(2\mu)}{Z_Q(\mu)^2}.$$

Inserting the previous computed values of these partition functions, we have

$$\begin{aligned}
W &= \frac{(\cosh 2\lambda + 1)(e^{2\mu}\cosh 2\lambda + e^{-2\mu})}{(e^\mu \cosh 2\lambda + e^{-\mu})^2} \\
&= \frac{e^{2\mu}\cosh^2 2\lambda + e^{-2\mu} + \cosh 2\lambda(e^{2\mu} + e^{-2\mu})}{(e^\mu \cosh 2\lambda + e^{-\mu})^2} \\
&= 1 + \frac{\cosh 2\lambda(e^\mu - e^{-\mu})^2}{(e^\mu \cosh 2\lambda + e^{-\mu})^2} \\
&\leq 1 + \frac{(e^\mu - e^{-\mu})^2}{e^{2\mu}\cosh 2\lambda} \\
&\leq 1 + \frac{4\sinh^2 \mu}{\cosh^2 \mu \cosh 2\lambda} \\
&\leq 1 + 8e^{-2\lambda}\tanh^2 \mu
\end{aligned}$$

where the second and third inequalities both use that $e^x \geq \cosh x \geq e^x/2$, for $x \geq 0$. $\qquad\square$

### F.1.2 Fan with deletions

In keeping with the rest of the text, these proofs will set $2B = d$. Note that the value of $B$ does not enter the resulting bounds.

*Proof of Proposition 23.* Let

$$P_{\ell,\eta,\mu,\lambda}(x) := \frac{1}{Z(\ell,\eta,\mu,\lambda)} \exp\left(\lambda x_{d+1}(\sum_{i=1}^{d/2} x_{2i}) + \mu x_{d+1}(\sum_{i=\ell+1}^{d/2} x_{2i-1})\right)$$

$$\cdot \exp\left(\eta x_{d+1}(\sum_{i=1}^{\ell} x_{2i-1}) + \lambda(\sum_{i=1}^{d/2} x_{2i}x_{2i-1})\right).$$

Then $P_{\ell,\mathrm{Fan}} = P_{\ell,0,\mu,\lambda}$, $Q_{\ell,\mathrm{Fan}} = P_{\ell,\mu,\mu,\lambda}$. Further, $Z_{2Q-P} = Z(\ell, 2\mu, \mu, \lambda)$.

Here again the partition function is simple to compute. In essence, the groups $(x_{2i-1}, x_{2i})$ across $i$ are independent given $x_{d+1}$, and the expressions, unsurprisingly, are invariant to value of $x_{d+1}$.

Unfortunately the calculations get a little messy. If one is not interested in the results on property testing in §E.2, then the following may be safely skipped. We do note that the steps below are elementary, it is just the form of the expressions that is long.

$$Z(\ell, \eta, \mu, \lambda)$$

$$= \sum_{x_{d+1}} \sum \exp\left(\lambda x_{d+1}(\sum_{i=1}^{d/2} x_{2i}) + \mu x_{d+1}(\sum_{i=\ell+1}^{d/2} x_{2i-1}) + \eta x_{d+1}(\sum_{i=1}^{\ell} x_{2i-1}) + \lambda(\sum_{i=1}^{d/2} x_{2i}x_{2i-1})\right)$$

$$= \sum_{x_{d+1}} \prod_{i=1}^{\ell} \sum_{x_{2i-1},x_{2i}} e^{x_{d+1}(\eta x_{2i-1} + \lambda x_{2i}) + \lambda x_{2i}x_{2i-1}} \cdot \prod_{i=\ell+1}^{d/2} \sum_{x_{2i-1},x_{2i}} e^{x_{d+1}(\mu x_{2i-1} + \lambda x_{2i}) + \lambda x_{2i}x_{2i-1}}$$

$$= \sum_{x_{d+1}} \left(2e^{\lambda}\cosh((\lambda+\eta)x_{d+1}) + 2e^{-\lambda}\cosh((\lambda-\eta)x_{d+1})\right)^{\ell}$$

$$\cdot \left(2e^{\lambda}\cosh((\lambda+\mu)x_{d+1}) + 2e^{-\lambda}\cosh((\lambda-\mu)x_{d+1})\right)^{d/2-\ell}$$

$$= 2^{d+1}\left(e^{\lambda}\cosh(\lambda+\eta) + e^{-\lambda}\cosh(\lambda-\eta)\right)^{\ell}\left(e^{\lambda}\cosh(\lambda+\mu) + e^{-\lambda}\cosh(\lambda-\mu)\right)^{d/2-\ell}$$

Thus,

$$1 + \chi^2(Q\|P) = \frac{Z(\ell,0,\mu,\lambda)Z(\ell,2\mu,\mu,\lambda)}{Z(\ell,\mu,\mu,\lambda)^2}$$

$$= \left(\frac{\left(e^{\lambda}\cosh(\lambda) + e^{-\lambda}\cosh(\lambda)\right)\left(e^{\lambda}\cosh(\lambda+2\mu) + e^{-\lambda}\cosh(\lambda-2\mu)\right)}{\left(e^{\lambda}\cosh(\lambda+\mu) + e^{-\lambda}\cosh(\lambda-\mu)\right)^2}\right)^{\ell}$$

$$=: U^{\ell}.$$

We proceed to estimate $U$.

$$U = \frac{\left(e^{\lambda}\cosh(\lambda) + e^{-\lambda}\cosh(\lambda)\right)\left(e^{\lambda}\cosh(\lambda+2\mu) + e^{-\lambda}\cosh(\lambda-2\mu)\right)}{\left(e^{\lambda}\cosh(\lambda+\mu) + e^{-\lambda}\cosh(\lambda-\mu)\right)^2}$$

$$= \frac{e^{2\lambda}\cosh\lambda\cosh(\lambda+2\mu) + e^{-2\lambda}\cosh\lambda\cosh(\lambda-2\mu) + \cosh(\lambda)\cosh(\lambda+2\mu) + \cosh(\lambda)\cosh(\lambda-2\mu)}{e^{2\lambda}\cosh^2(\lambda+\mu) + e^{-2\lambda}\cosh^2(\lambda-\mu) + 2\cosh(\lambda+\mu)\cosh(\lambda-\mu)}$$

By eliminating one factor of the denominator from the numerator above, we obtain the sequence of relations that follows below.

$$U \overset{(a)}{=} 1 + \frac{(e^{2\lambda} + e^{-2\lambda})\sinh^2\mu + \sinh(\mu)\left(\sinh(2\lambda+\mu) - \sinh(2\lambda-\mu)\right)}{e^{2\lambda}\cosh^2(\lambda+\mu) + e^{-2\lambda}\cosh^2(\lambda-\mu) + 2\cosh(\lambda+\mu)\cosh(\lambda-\mu)}$$

$$\overset{(b)}{=} 1 + \frac{2\cosh(2\lambda)\sinh^2\mu + 2\cosh(2\lambda)\sinh^2\mu}{\left(e^{\lambda}\cosh(\lambda+\mu) + e^{-\lambda}\cosh(\lambda-\mu)\right)^2}$$

$$= 1 + \frac{4\sinh^2(\mu)\cosh(2\lambda)}{e^{2\lambda}\cosh^2(\lambda+\mu) + e^{-2\lambda}\cosh^2(\lambda-\mu) + 2\cosh(\lambda+\mu)\cosh(\lambda-\mu)}$$

$$\overset{(c)}{\leq} 1 + 4\frac{\sinh^2\mu}{\cosh^2(\lambda+\mu)} \leq 1 + 4\frac{\sinh^2\mu}{\cosh^2\lambda\cosh^2\mu}$$

$$\leq 1 + 16e^{-2\lambda}\tanh^2\mu,$$

where $(a)$ follows by the identities

$$\cosh(u)\cosh(u+2v) - \cosh^2(u+v) = \sinh^2 v$$

$$\cosh(u)\cosh(u+2v) - \cosh(u+v)\cosh(u-v) = \sinh(v)\sinh(2u+v),$$

$(b)$ uses

$$\sinh(2u+v) - \sinh(2u-v) = 2\cosh(2u)\sinh u,$$

and $(c)$ follows by dropping all terms but the first in the denominator, and observing that $e^{2\lambda} \geq \cosh(2\lambda)$. Finally, the inequality $\cosh(\lambda+\mu) \geq \cosh\lambda\cosh\mu$ holds because $\lambda, \mu \geq 0$.

$\square$

## F.2 Clique-based Widgets

The method for showing the bounds is developed in the case of the Clique with a single edge deleted. While there are variations in the proofs of the following two cases, the basic recipe remains the same.

We begin with a technical lemma that is repeatedly used in the following.

**Lemma 30.** *Let $\tau : [a, b] \to \mathbb{R}$ be a function differentiable on $(a, b)$ such that $\tau'$ is strictly concave. If $\tau(a) < 0$ and $\tau(b) > 0$, then $\tau$ has exactly one root in $(a, b)$*

*Proof.* Since $\tau'$ is concave, it can have at most two roots in $(a, b)$. Indeed, if there were three roots $a < x_1 < x_2 < x_3 < b$, then $\exists t \in (0, 1) : x_2 = tx_1 + (1 - t)x_3$, and $0 = f(x_2) = tf(x_1) + (1 - t)f(x_3)$ violates strict concavity. Further, between its roots, $\tau'$ must be positive, again by concavity.

Thus, we can break $[a, b]$ into three intervals $(I_1, I_2, I_3)$, some of them possibly trivial[8], of the from $([a, x_1), [x_1, x_2], (x_2, b])$, such that $\tau$ is monotone decreasing on the interiors of $I_1, I_3$ and monotone increasing on the interior of $I_2$.

Note that $\tau$ has at least one root by the intermediate value theorem. We now argue that it cannot have more than one. Since $\tau$ is falling on $I_1$, it follows that $\sup_{x \in I_1} \tau(x) = \tau(a) < 0$, and there is no root in $I_1$. Similarly, since $\tau$ is falling on $I_3$, $\tau(b) = \inf_{x \in I_3} \tau(x) > 0$, and there is no root in $I_3$. This leaves $I_2$, and since $\tau$ is monotone on $I_2$, it has at most one root on the same. $\qquad\square$

### F.2.1 Clique with a single edge deleted

*Proof of Proposition 25.* Let $P = P_{\text{Clique}}$ and $Q = Q_{\text{Clique}}$ as defined in the main text. For given $\lambda, \eta$, let

$$P_{\lambda,\eta}(x) := \frac{1}{Z(\lambda, \eta)} e^{\frac{\lambda}{2}\left((\sum x_i)^2 - (d+1)\right)} e^{-\eta x_1 x_2}$$

Note that $P = P_{\lambda, \lambda - \mu}$, and $Q = P_{\lambda, \lambda}$. Further,

$$W := \mathbb{E}_P[(Q/P)^2] = \frac{Z(\lambda, \lambda - \mu)Z(\lambda, \lambda + \mu)}{Z(\lambda, \lambda)^2}.$$

We begin by writing $Z$ in a convenient form, derived by breaking the configurations into bins depending on the number of $x_i$s that take the value $-1$:

$$Z(\lambda, \eta) = \sum_{j=0}^{d-1} \binom{d-1}{j} \left\{ e^{-\eta} \left( e^{\frac{\lambda}{2}(d+1-2j)^2 - (d+1)} + e^{\frac{\lambda}{2}(d-3-2j)^2 - (d+1)} \right) + 2e^{\eta} e^{\frac{\lambda}{2}((d-1-2j)^2 - (d+1))} \right\}.$$

Notice above that since $(d - 3 - 2(d - 1 - j))^2 = (d + 1 - 2j)^2$, and $\binom{d-1}{j} = \binom{d-1}{d-1-j}$, it follows that the sums over the first two terms above are identical. Thus,

$$Z(\lambda, \eta) = 2\sum \binom{d-1}{j} e^{-\eta} e^{\frac{\lambda}{2}(d+1-2j)^2 - (d+1)} + 2\sum e^{\eta} e^{\frac{\lambda}{2}((d-1-2j)^2 - (d+1))}$$

$$\iff \underbrace{\frac{Z(\lambda, \eta)}{2e^{\lambda/2(d^2 - d)}}}_{=:\widetilde{Z}(\lambda,\eta)} = e^{\lambda d - \eta} \underbrace{\sum \binom{d-1}{j} e^{-2\lambda j(d+1-j)}}_{=:S_1(\lambda)} + e^{-(\lambda d - \eta)} \underbrace{\sum \binom{d-1}{j} e^{-2\lambda j(d-1-j)}}_{=:S_2(\lambda)}$$

$$\iff \widetilde{Z}(\lambda, \eta) = e^{\lambda d - \eta} S_1(\lambda) + e^{-\lambda d + \eta} S_2(\lambda).$$

Since the term appears often, we set $d' = d - 1$. As a consequence of the above, we have

$$W = \frac{Z(\lambda, \lambda - \mu)Z(\lambda, \lambda + \mu)}{Z(\lambda, \lambda)^2} = \frac{\widetilde{Z}(\lambda, \lambda - \mu)\widetilde{Z}(\lambda, \lambda + \mu)}{\widetilde{Z}(\lambda, \lambda)^2}$$

$$= \frac{(e^{\lambda d' + \mu} S_1(\lambda) + e^{-\lambda d' - \mu} S_2(\lambda))(e^{\lambda d' - \mu} S_1(\lambda) + e^{-\lambda d' + \mu} S_2(\lambda))}{(e^{\lambda d'} S_1(\lambda) + e^{-\lambda d'} S_2(\lambda))^2}$$

$$= 1 + 4 \sinh^2 \mu \frac{S_1 S_2}{(e^{\lambda d'} S_1 + e^{-\lambda d'} S_2)^2}$$

$$\leq 1 + 4 \sinh^2 \mu \frac{e^{-2\lambda d'} S_2(\lambda)}{S_1(\lambda)}.$$

The bounds are now forthcoming by controlling $S_1, S_2$ as in the following

**Lemma 31.** *If $d \geq 5$ and $\lambda(d - 4) \geq \log(d)$, then*

$$S_1(\lambda) \geq 1$$
$$S_2(\lambda) \leq 2 + 3de^{-2\lambda(d-2)} \leq 2 + 3/d.$$

The bound follows directly from the control offered above. $\qquad\square$

This proof describes closely the structure of the forthcoming proofs

- Begin by introducing one free parameter, $\eta$ varying which yields Ising models that interpolate between $P$ and $Q$.
- Express the $\chi^2$ divergence as a ratio of partition functions.
- Exploit the symetries of the mean field Ising model to more conveniently write these partition functions.
- Control the terms arising via a 'ratio trick' as in the proof of Lemma 31. At time this is used more than once, or a more direct form of this trick is used instead.

We conclude by showing Lemma 31.

*Proof of Lemma 31.* $S_1 \geq 1$ follows trivially, since all terms in the sum are non-negative and the first term is $\binom{d-1}{0}e^0 = 1$.

Concentrating on $S_2$, let $T_j := \binom{d-1}{j}e^{-2\lambda j(d-1-j)}$. Note that $S_2 = \sum T_j$, and that $T_j = T_{d-1-j}$ for every $j$. Further, for $j \in [0 : d-2]$,

$$\frac{T_{j+1}}{T_j} = \frac{d-1-j}{j+1}e^{-2\lambda(d-2-2j)}.$$

Treating $j$ as a real number in $[0, d-2]$, define

$$\tau(j) = \log(d-1-j) - \log(j+1) - 2\lambda(d-2-2j).$$

We have

$$\tau'(j) = -\frac{1}{d-1-j} - \frac{1}{j+1} + 4\lambda$$

$$\tau''(j) = -\frac{1}{(d-1-j)^2} + \frac{1}{(j+1)^2}$$

$$\tau'''(j) = -\frac{2}{(d-1-j)^3} - \frac{2}{(j+1)^3} < 0.$$

We may now note that $\tau'$ is a strictly concave function on the relevant domain. Further, note that since $\log(d-1) \leq 2\lambda(d-2)$ follows from our conditions, $\tau(0) < 0$, and similarly, $\tau(d-2) > 0$.

By Lemma F.2, $\tau$ has exactly one root in $[0, d-2]$ - in particular, this lies at $j = d/2 - 1$. But since $T_{j+1}/T_j = e^{\tau(j)}$, we obtain that for $j \leq d/2 - 1, T_{j+1} \leq T_j$, and for $j \geq d/2 - 1, T_{j+1} \geq T_j$.

Since $T$s are decreasing until $d/2 - 1$ and increasing after $d/2$, it follows that for all $j \in [2 : d-3]$, $T_j \leq \max(T_2, T_{d-3}) = T_2$. Now, under the conditions of the theorem,

$$
\begin{aligned}
\frac{T_2}{T_1} &= \exp(\tau(1)) = \exp(\log(d-2) - \log 2 - 2\lambda(d-4)) \\
&\leq \exp(\log(d-2) - \log 2 - 2\log(d)) \leq 1/d,
\end{aligned}
$$

where we have used the assumption $\lambda(d-4) \geq \log d$. Thus,

$$
\begin{aligned}
S_2 &= T_0 + T_1 + \sum_{j=2}^{d-3} T_j + T_{d-2} + T_{d-1} \\
&\leq 1 + T_1 + \frac{d-4}{d}T_1 + T_1 + 1 \\
&\leq 2 + 3d\exp(-2\lambda(d-2)) \leq 2 + 3/d. \qquad \square
\end{aligned}
$$

We call this method of estimating sums such as $S_2$ the *ratio trick*, since they control the values of the sums by controlling the ratios of subsequent terms.

### F.2.2 Clique with Large Hole

The computations of this section are in essence a deepening of the previous section, and we will frequently make references to the same.

*Proof of Proposition 26.* Once again condensing notation, let $P := P_{\ell,\mathrm{Clique}}, Q := Q_{\ell,\mathrm{Clique}}$.

Further, let

$$
P_{\ell,\lambda,\eta}(x) := \frac{1}{Z_\ell(\lambda,\eta)} e^{\frac{\lambda}{2}\left(\sum_{1 \leq i \leq d+1} x_i\right)^2 - (d+1)} e^{-\frac{\eta}{2}\left(\sum_{1 \leq i \leq \ell} x_i\right)^2 - \ell}
$$

Again, $P = P_{\ell,\lambda,\lambda-\mu}, Q = P_{\ell,\lambda,\lambda}$ holds. $Z_\ell$ is the central object for this section, and has the following expression. This is derived by tracking the number of negative $x_i$s in both the bulk of the clique and the single 'hole' separately.

$$
\begin{aligned}
Z_\ell(\lambda,\eta) &:= \sum_{\{\pm 1\}^{d+1}} e^{\frac{\lambda}{2}\left(\sum_{1 \leq i \leq d+1} x_i\right)^2 - (d+1)} e^{-\frac{\eta}{2}\left(\sum_{1 \leq i \leq \ell} x_i\right)^2 - \ell} \\
&= \sum_{i,j} \binom{\ell}{i}\binom{d+1-\ell}{j} e^{\frac{\lambda}{2}(d+1-2i-2j)^2 - (d+1)} e^{\frac{-\eta}{2}(\ell-2i)^2 - \ell}
\end{aligned}
$$

We normalise $Z_\ell$ by $e^{\lambda/2((d+1)^2 - (d+1))} e^{-\eta/2(\ell^2 - \ell)}$, and put a $\sim$ over the normalised version[9] to get

$$
\begin{aligned}
\widetilde{Z}_\ell(\lambda,\eta) &:= \sum_{i,j} \binom{\ell}{i}\binom{d+1-\ell}{j} e^{-2\lambda j(d+1-2i-j)} e^{2\eta i(\ell-i)} e^{-2\lambda i(d+1-i)} \\
&=: \sum_{i=0}^{\ell} \binom{\ell}{i} e^{2\eta i(\ell-i)} e^{-2\lambda i(d+1-i)} S_i(\lambda)
\end{aligned}
$$

where

$$S_i(\lambda) := \sum_j \binom{d+1-\ell}{j} e^{-2\lambda j(d+1-2i-j)}.$$

Notice that $S_i \geq 0$ for every $i$.

As before, we are interested in controlling

$$W := \frac{Z_\ell(\lambda, \lambda - \mu) Z_\ell(\lambda, \lambda + \mu)}{Z_\ell(\lambda, \lambda)^2} = \frac{\widetilde{Z}_\ell(\lambda, \lambda - \mu) \widetilde{Z}_\ell(\lambda, \lambda + \mu)}{\widetilde{Z}_\ell(\lambda, \lambda)^2}.$$

To this end, note first that $2\lambda i(\ell - i) - 2\lambda i(d+1-i) = -2\lambda i(d+1-\ell)$, and so, for instance,

$$\widetilde{Z}_\ell(\lambda, \lambda + \mu) = \sum_i \binom{\ell}{i} e^{2\mu i(\ell - i)} e^{-2\lambda i(d+1-\ell)} S_i(\lambda).$$

Collecting like terms in expressions of the above form, we obtain that

$$\frac{\widetilde{Z}_\ell(\lambda, \lambda - \mu)}{\widetilde{Z}_\ell(\lambda, \lambda)} = 1 + \frac{\sum_{i=1}^{\ell-1} \binom{\ell}{i} \left( e^{-2\mu i(\ell-i)} - 1 \right) e^{-2\lambda i(d+1-\ell)} S_i(\lambda)}{\widetilde{Z}_\ell(\lambda, \lambda)}$$

and

$$\frac{\widetilde{Z}_\ell(\lambda, \lambda + \mu)}{\widetilde{Z}_\ell(\lambda, \lambda)} = 1 + \frac{\sum_{i=1}^{\ell-1} \binom{\ell}{i} \left( e^{2\mu i(\ell-i)} - 1 \right) e^{-2\lambda i(d+1-\ell)} S_i(\lambda)}{\widetilde{Z}_\ell(\lambda, \lambda)},$$

where the terms involving $i = 0$ and $i = \ell$ in the numerator drop out because $e^{2\mu i(\ell - i)} = 1$ in these cases.

Now, if $\mu \geq 0$ the second terms in the above two expressions are respectively negative and positive, while if $\mu < 0$, they are respectively positive and negative. It is a triviality that for $A < 0 < B, (1 + A)(1 + B) \leq 1 + A + B$. We thus have the upper bound

$$W \leq 1 + \frac{\sum_{i=1}^{\ell-1} \binom{\ell}{i} 2 \left( \cosh 2\mu i(\ell - i) - 1 \right) e^{-2\lambda i(d+1-\ell)} S_i(\lambda)}{\widetilde{Z}_\ell(\lambda, \lambda)}$$

$$= 1 + 4 \frac{\sum_{i=1}^{\ell-1} \binom{\ell}{i} \sinh^2(\mu i(\ell - i)) e^{-2\lambda i(d+1-\ell)} S_i(\lambda)}{\widetilde{Z}_\ell(\lambda, \lambda)} \qquad (1)$$

While we will provide full proofs in the sequel, it may help to see where we are going first. Roughly, we argue via the ratio trick in the proof of Lemma 31 in the previous section, that $S_i$ is bounded by $2(1 + e^{-2\lambda(\ell-2i)(d+1-\ell)})$, under conditions such as $\lambda(d + 1 - 2\ell) \geq \log d + 1 - 2\ell$. Plugging in this upper bound, and noting that after multiplication with $e^{-2\lambda i(d+1-\ell)}$ we have a sum that is completely symmetric under $i \mapsto \ell - i$, we can bound $W$ as

$$W \leq 1 + 16 \frac{\sum_{i=1}^{\ell-1} \binom{\ell}{i} \sinh^2(\mu i(\ell - i)) e^{-2\lambda i(d+1-\ell)}}{\widetilde{Z}_\ell(\lambda, \lambda)}.$$

We then show that under the conditions of the proposition, the first term in the above sum dominates all the remaining terms, in the process utilising the condition $|\mu| \leq \lambda$. Finally, using the trivial bound $\widetilde{Z}_\ell(\lambda, \lambda) \geq 1$, we get the claied upper bound.

Let us then proceed. The control on the $S_i$s is offered below.

**Lemma 32.** *If $\lambda(d + 1 - 2\ell) \geq \log(d + 1 - 2\ell)$ and $d \geq 4\ell$, then for every $i \in [1 : \ell - 1]$,*

$$S_i(\lambda) \leq 2 + 2e^{-2\lambda(\ell-2i)(d+1-\ell)}.$$

Incorporating the above lemma into (1), we have

$$W \leq 1 + 8 \frac{\sum_{i=1}^{\ell-1} \binom{\ell}{i} \sinh^2(\mu i(\ell-i)) e^{-2\lambda i(d+1-\ell)} \left(1 + e^{-2\lambda(\ell-2i)(d+1-\ell)}\right)}{\widetilde{Z}_\ell(\lambda, \lambda)}$$

$$\leq 1 + 8 \frac{\sum_{i=1}^{\ell-1} \binom{\ell}{i} \sinh^2(\mu i(\ell-i)) \left(e^{-2\lambda i(d+1-\ell)} + e^{-2\lambda(\ell-i)(d+1-\ell)}\right)}{\widetilde{Z}_\ell(\lambda, \lambda)}$$

$$\stackrel{(a)}{=} 1 + 16 \frac{\sum_{i=1}^{\ell-1} \binom{\ell}{i} \sinh^2(\mu i(\ell-i)) e^{-2\lambda i(d+1-\ell)}}{\widetilde{Z}_\ell(\lambda, \lambda)}$$

$$= 1 + \frac{16}{\widetilde{Z}_\ell(\lambda, \lambda)} \left( \sinh^2(\mu(\ell-1)) e^{-2\lambda(d+1-\ell)} + \sum_{i=2}^{\ell-1} \binom{\ell}{i} \sinh^2(\mu i(\ell-i)) e^{-2\lambda i(d+1-\ell)} \right)$$

$$\stackrel{(b)}{\leq} 1 + \frac{16}{\widetilde{Z}_\ell(\lambda, \lambda)} \left( \sinh^2(\mu(\ell-1)) e^{-2\lambda(d+1-\ell)} + \sum_{i=2}^{\ell-1} \binom{\ell}{i} e^{2|\mu| i\ell - 2\lambda i(d+1-\ell)} \right)$$

$$\stackrel{(c)}{\leq} 1 + \frac{16}{\widetilde{Z}_\ell(\lambda, \lambda)} \left( \sinh^2(\mu(\ell-1)) e^{-2\lambda(d+1-\ell)} + \sum_{i=2}^{\ell-1} \binom{\ell}{i} e^{-2\lambda i(d+1-2\ell)} \right) \qquad (2)$$

where the equality $(a)$ follows since each term in the sum is invariant under the map $i \mapsto \ell - i$, $(b)$ follows since $\sinh x \leq e^x$, and $(c)$ used $\lambda \geq |\mu|$. .

For $i \in [2 : \ell]$, let $V_i$ denote the term corresponding to $i$ in the summation above, and let $V_1 = \sinh^2(\mu(\ell-1) e^{-2\lambda(d+1-\ell)}$. We will argue that $V_1$ dominates $V_i$ for every $i$ by using a weakened ratio trick.

Note that

$$V_1 \geq e^{-2\lambda(d+1-\ell)-2|\mu|(\ell-1)} \geq e^{-2\lambda d}.$$

Further,

$$\frac{V_i}{V_1} \leq \exp\left(i \log \ell + 2\lambda d - 2\lambda i(d+1-2\ell)\right).$$

This is smaller than $1/\ell$ so long as for every $i$,

$$i(2\lambda(d+1-2\ell) - \log \ell) > 2\lambda d + \log(\ell),$$

which hold if the following conditions are true:

$$2\lambda(d+1-2\ell) > \log \ell$$
$$4\lambda(d+1-2\ell) > 3\log \ell + 2\lambda d.$$

The above hold if $\lambda(d+2-4\ell) \geq 3/2 \log \ell$, which is true under the conditions of the proposition since $\ell < d/8$, and since $\lambda(d+2-4\ell) \geq \lambda d/2 \geq 3/2 \log d$.

Finally, it remains to show that $\widetilde{Z}_\ell(\lambda, \lambda)$ is non-trivially large. But note that $\widetilde{Z}_\ell(\lambda, \lambda) \geq S_0(\lambda) \geq 1$.

Thus, we have shown that

$$W \leq 1 + 32\ell \sinh^2(\mu(\ell-1)) e^{-2\lambda(d+1-\ell)}.$$

$\square$

*Proof of Lemma 32.* For $j \in [0 : d+1-\ell]$, let

$$T_j := \binom{d+1-\ell}{j} e^{-2\lambda j(d+1-2i-j)}.$$

Recall that $S_i = \sum T_j$. We will use the ratio trick again. To this end, observe that

$$\frac{T_{j+1}}{T_j} = \frac{d+1-\ell-j}{j+1} \exp\left(-2\lambda(d-2i-2j)\right).$$

Again treating $j$ as a real number in $[0 : d - \ell]$, let
$$\tau(j) := \log(d + 1 - \ell - j) - \log(1 + j) - 2\lambda(d - 2i - 2j).$$
By considerations similar to the previous section, $\tau$ is strictly concave, and by Lemma F.2, $\tau$ has exactly one root so long as $\tau(0) < 0$ and $\tau(d - \ell) > 0$. In this setting these conditions translate to
$$\log(d + 1 - \ell) < 2\lambda(d - 2i)$$
$$\log(d + 1 - \ell) < -2\lambda(d - 2i - 2(d - \ell)) = 2\lambda(d - 2(\ell - i)).$$

The above hold for every $i$ so long as $\log(d + 1 - \ell) < 2\lambda(d + 2 - 2\ell)$.

Since $\tau$ has a single root and is initially negative, we again find that for all $j \in [2 : d - 1 - \ell]$, $T_j \le \max(T_2, T_{d-1-\ell})$. Further,
$$\frac{T_2}{T_1} = \frac{d - \ell}{2} \exp(-2\lambda(d - 2 - 2i)) \le \frac{d - \ell}{2} \exp(-2\lambda(d - 2\ell)) \le \frac{1}{d - \ell}$$
$$\frac{T_{d-\ell-1}}{T_{d-\ell}} = \frac{d - \ell}{2} \exp(-2\lambda(d - 2(\ell - i)) \le \frac{1}{d - \ell}.$$

Further,
$$\max\left(\frac{T_1}{T_0}, \frac{T_{d-\ell}}{T_{d+1-\ell}}\right) \le (d + 1 - \ell)e^{-2\lambda(d-2\ell)} \le 1/2.$$

Thus,
$$S_1 \le T_0 + T_{d+1-\ell} + (1 + (d - \ell - 2)/(d - \ell)) \max(T_1, T_{d-\ell})$$
$$\le T_0 + T_{d+1-\ell} + 2 \max(T_1, T_{d-\ell})$$
$$\le 2(T_0 + T_{d+1-\ell}).$$

Now notice that
$$T_0 = 1$$
$$T_{d-\ell+1} = \exp(-2\lambda(d + 1 - \ell)(d + 1 - 2i - d - 1 + \ell)) = \exp(-2\lambda(\ell - 2i)(d + 1 - \ell)),$$
and thus the claim follows. $\qquad\square$

We now prove the reverse direction, i.e. control on $\chi^2(P\|Q)$. This is essentially a small variation on the previous setting.

*Proof of Proposition 27.* Referring to the previous proof, we instead need to control
$$W' = \frac{\widetilde{Z}_\ell(\lambda, \lambda)\widetilde{Z}_\ell(\lambda, \lambda + 2\mu)}{\widetilde{Z}_\ell(\lambda, \lambda + \mu)^2}.$$
Proceeding in the same way, we may conntrol
$$W' \le 1 + \frac{\sum_{i=1}^{\ell-1}\binom{\ell}{i}\left(\cosh(4\mu i(\ell - i)) - 2\cosh(2\mu(i(\ell - i)) + 1\right)e^{-2\lambda i(d+1-\ell)}S_i(\lambda)}{\widetilde{Z}_\ell(\lambda, \lambda + \mu)}$$

For succinctness, let $f(x) := \cosh(4\mu x) - 2\cosh(2\mu x) + 1$. Note that $1 \le f(x) \le e^{4|\mu||x|}$. Since the $S_i$ are identical to the previous case, Lemma 32 applies, and
$$W' \le 1 + 8\frac{\sum_{i=1}^{\ell-1}\binom{\ell}{i}f(i(\ell - i))e^{-2\lambda i(d+1-\ell)}\left(1 + e^{-2\lambda(\ell - 2i)(d+1-\ell)}\right)}{\widetilde{Z}_\ell(\lambda, \lambda + \mu)}$$
$$\le 1 + 16\frac{\sum_{i=1}^{\ell-1}\binom{\ell}{i}f(i(\ell - i))e^{-2\lambda i(d+1-\ell)}}{\widetilde{Z}_\ell(\lambda, \lambda + \mu)}$$
$$\le 1 + \frac{16}{\widetilde{Z}_\ell(\lambda, \lambda + \mu)}\left(f(\ell - 1)e^{-2\lambda(d+1-\ell)} + \sum_{i=2}^{\ell-1}\binom{\ell}{i}e^{4|\mu|i\ell - 2\lambda i(d+1-\ell)}\right)$$
$$\le 1 + \frac{16}{\widetilde{Z}_\ell(\lambda, \lambda + \mu)}\left(f(\ell - 1)e^{-2\lambda(d+1-\ell)} + \sum_{i=2}^{\ell-1}\binom{\ell}{i}e^{-2\lambda i(d+1-3\ell)}\right)$$

Notice the distinction that the exponent in the second sum contains a $-3\ell$ instead of a $-2\ell$. Using $f(x) \geq 1$, the same control on the relative values of $S_i$ and the summation holds as long as

$$4\lambda(d + 1 - 3\ell) > 3\log\ell + 2\lambda d.$$

This translates to demanding that $2\lambda(d-6\ell) > 3/2\lambda d$, which holds for $\ell \leq d/12$. Finally, $\widetilde{Z}_\ell(\lambda, \lambda + \mu) \geq 1$ as well, and thus,

$$W' \leq 1 + 32\ell e^{-2\lambda(d+1-\ell)}\left(\cosh(4\mu(\ell-1)) - 2\cosh(2\mu(\ell-1)) + 1\right).$$

Finally, we note that for any $x$,

$$
\begin{aligned}
\cosh(4x) - 2\cosh(2x) + 1 &= \sinh^2(2x) + (\cosh(2x) - 1)^2 \\
&= 4\sinh^2 x \cosh^2 x + 4\sinh^4 x = 4\sinh^2 x \cosh^2 x(1 + \tanh^2 x) \\
&\leq 2\sinh^2(2x). \qquad \square
\end{aligned}
$$

### F.2.3 Emmentaler Cliques

*Proof of Proposition 28.* Recall the setup - $d + 1$ nodes are divided into $B = d/(\ell+1)$ groups of $\ell + 1$ nodes each, denoted $V_1, \ldots, V_B$, and the final node $d + 1$ is kept separate. Recall that for a set $S$, $x_S := \sum_{u \in S} x_u$. Define

$$P_{\ell,\lambda,\eta} = \frac{1}{Z_\ell(\lambda,\eta)}\exp\left(\lambda/2\left(\sum_{i=1}^B x_{V_i}\right)^2 - \lambda/2\sum_{i=1}^B (x_{V_i}^2) + \lambda x_v \sum_{i=2}^B x_{V_i} + \eta x_v x_{V_1}\right).$$

Then $P = P_{\ell,\text{Emmentaler}} = P_{\ell,\lambda,0}, Q = Q_{\ell,\text{Emmentaler}} = P_{\ell,\lambda,\mu}$ and $Z_{2Q-P} = Z_\ell(\lambda, 2\mu)$ Marginalising over $x_v$, we get

$$Z_\ell(\lambda,\eta) = 2\sum_x \exp\left(\lambda/2\left(\sum_{i=1}^B x_{V_i}\right)^2 - \lambda/2\sum_{i=1}^B(x_{V_i}^2)\right)\cosh\left(\lambda\sum_{i=2}^B x_{V_i} + \eta x_{V_1}\right)$$

$$\leq 2\cosh(\lambda(d - \ell - 1) + \eta(\ell+1))\sum_x \exp\left(\lambda/2\left(\sum_{i=1}^B x_{V_i}\right)^2 - \lambda/2\sum_{i=1}^B(x_{V_i}^2)\right),$$

while dropping all terms for which $|\sum_i x_{V_i}| < d$, we get

$$
\begin{aligned}
Z_\ell(\lambda,\eta) &\geq 4\cosh(\lambda(d - \ell - 1) + \eta(\ell+1))e^{\lambda/2(B^2-B)(\ell+1)^2} \\
&= 4\cosh(\lambda(d' - \ell - 1) + \mu(\ell+1))e^{\lambda/2(d^2-d(\ell+1))}.
\end{aligned}
$$

To control $Z_\ell$ from above, it is necessary to control the partition function of the Emmentaler graph on $d$ nodes (i.e., with only the groups $V_1, \ldots V_B$, and without the extra node from above. We set this equal to $Y_\ell(\lambda)$. Then, similarly tracking configurations by the number of negative $x_i$s in each part,

$$
\begin{aligned}
Y_\ell &:= \sum_x \exp\left(\lambda/2\left(\sum_{i=1}^B x_{V_i}\right)^2 - \lambda/2\sum_{i=1}^B(x_{V_i}^2)\right). \\
&= \sum_{j_1,\ldots,j_B}\prod\binom{\ell+1}{j_i}\cdot\exp\left(\lambda/2\left((d - 2\sum j_i)^2 - \sum(\ell+1-2j_i)^2\right)\right) \\
&= e^{\lambda/2(d^2-d(\ell+1))}\sum_{j_1,\ldots,j_B}\prod\binom{\ell+1}{j_i}\cdot\exp\left(-2\lambda\left((d - \ell - 1)(\sum j_i) + \sum j_i^2 - (\sum j_i)^2\right)\right)
\end{aligned}
$$

For succinctness, let $d' := d - \ell - 1$. We establish the following lemma after concluding this argument

**Lemma 33.** *If $\ell \leq d/4$ and $\lambda(d-4) \geq 3\log(d)$, then*

$$Y_\ell \leq 2e^{\lambda/2(d^2 - d(\ell+1))}\left(1 + 2de^{-2\lambda d'}\right)$$

Invoking the above lemma and the previously argued control on $Z_\ell$, we get that

$$
\begin{aligned}
W := \mathbb{E}_P[(Q/P)^2] &= \frac{Z_\ell(\lambda, 0)Z_\ell(\lambda, 2\mu)}{Z_\ell(\lambda, \mu)^2} \\
&\leq \frac{\cosh(\lambda d')\cosh(\lambda d' + 2\mu(\ell+1))}{\cosh^2(\lambda d' + \mu(\ell+1))}\left(\frac{2Y_\ell}{4e^{\lambda/2(d^2-d(\ell+1))}}\right)^2 \\
&\leq \left(1 + \frac{\sinh^2(\mu(\ell+1))}{\cosh^2(\lambda d' + \mu(\ell+1))}\right)\left(1 + 2de^{-2\lambda d'}\right)^2 \\
&\leq \left(1 + 4\tanh^2(\mu(\ell+1))e^{-2\lambda d'}\right)\left(1 + 2de^{-2\lambda d'}\right)^2
\end{aligned}
$$

Under the conditions of the theorem, both $4\tanh^2(\mu(\ell+1))e^{-2\lambda d'}$ and $2de^{-2\lambda d'}$ are smaller than $1/4$. But for $x, y$, it holds that $(1+x)^2 < 1+3x$ and $(1+3x)(1+y) < 1+4(x+y) \leq 1+8\max(x,y)$. Lastly, $4\tanh^2 x \leq 4 \leq d$, and thus, we have shown the bound

$$W \leq 1 + 32de^{-2\lambda(d-\ell-1)}. \qquad \square$$

*Proof of Lemma 33.* Fix a vector $(j_1, \ldots, j_B)$ and let $k := \sum j_i$. We will argue the claim by controlling the terms in $Y_\ell$ with a given value of $k$.

**Lemma 34.** *If $\sum j_i = k \in [2 : d-2]$, $\ell + 1 \leq d/4$ and $\lambda(d-4) \geq 3\log(d)$, then*

$$\prod\binom{\ell+1}{j_i} \cdot \exp\left(-2\lambda\left(d'(\sum j_i) + \sum j_i^2 - (\sum j_i)^2\right)\right) \leq \frac{1}{d^{\min(k, d-k)}}e^{-2\lambda d'}.$$

Thus, we have the bound

$$\frac{Y_\ell}{e^{\lambda/2(d^2 - d(\ell+1))}} \leq 2\left(1 + B(\ell+1)e^{-2\lambda d'}\right) + \sum_{k=2}^{d-2}\frac{N_k}{d^{\min(k, d-k)}}e^{-2\lambda d'},$$

where

$$N_k = \left|\left\{j \in [0 : \ell+1]^B : \sum j_i = k\right\}.\right|$$

Notice that $N_k = N_{d-k}$. Further, for $k \leq d/2$, by stars and bars,

$$N_k \leq \binom{k+B-1}{k} \leq (1 + (B-1)/k)^{k-1} \leq B^k \leq d^k$$

Consequently, $N_k \leq d^{\min(k, d-k)}$, and we have established the upper bound

$$\frac{Y_\ell}{2e^{\lambda/2(d^2 - d(\ell+1))}} \leq 1 + 2de^{-2\lambda d'}. \qquad \square$$

*Proof of Lemma 34.* Note that $\binom{n}{m} \leq n^{\min(m, n-m)}$. Therefore,

$$\prod\binom{\ell+1}{j_i} \leq \exp\left(\min(k, d-k)\log(\ell+1)\right).$$

Next, by Cauchy-Schwarz,

$$\sum j_i^2 \geq \frac{(\sum j_i)^2}{B} = k^2\left(1 - \frac{d'}{d}\right).$$

Let LHS, RHS be the left and right hand sides of the inequality claimed in the Lemma. Using the above,

$$\log \frac{\text{LHS}}{\text{RHS}} \leq \min(k, d-k) \log(d(\ell+1)) - 2\lambda \left( d'k + k^2 d'/d - d' \right)$$

$$= \min(k, d-k) \log(d(\ell+1)) - 2\lambda \frac{d'}{d} \left( k(d-k) - d \right).$$

Let $f(k)$ be the upper bound above. Notice that $f(k) = f(d-k)$. Thus, it suffices to show that $f(u) \leq 0$ for every real number $u \in [2, d/2]$.

For a real number $u \in [2, d/2)$, it holds that $f''(u) = 4\lambda > 0$. It follows that $f$ attains its maxima on $\{2, d/2\}$. Since $\ell + 1 < d/4$, we have $d'/d \geq 3/4$, and thus

$$f(2) = 2\log(d(\ell+1)) - 2\lambda \frac{d'}{d}(d-4) \leq 4\log(d) - \frac{3}{2}\lambda(d-4) < 0$$

$$f(d/2) = \frac{d}{2}\left( \log(d(\ell+1) - 2\lambda\frac{d'}{d} \cdot \frac{(d-4)}{2} \right) = \frac{d}{4}f(2) < 0. \qquad \square$$

### F.2.4 Emmentaler v/s Full Clique

*Proof of Proposition 29.* Let

$$P_{\ell,\lambda,\eta}(x) := \frac{1}{Z_\ell(\lambda,\eta)} \exp\left( \lambda/2 \left( \left( \sum_{i=1}^{B} x_{V_i} \right)^2 - d \right) - (\lambda - \eta)/2 \sum_{i=1}^{B} (x_{V_i}^2 - (\ell+1)) \right).$$

Then $P_\ell = P_{\ell,\lambda,0}, Q_\ell = P_{\ell,\lambda,\mu}$. Let $d' = d - 1 - \ell$. Developing this a little, one can write

$$Z_\ell(\lambda,\eta) = C_{\ell,\lambda,\eta} \sum_{j_1,\dots,j_B} \prod \binom{\ell+1}{j_i} \cdot e^{-2\lambda\left( d' \sum j_i + \sum j_i^2 - (\sum j_i)^2 \right) - 2\eta\left( (\ell+1)\sum j_i - \sum j_i^2 \right)},$$

where

$$C_{\ell,\lambda,\eta} = \exp\left( \lambda/2(d^2 - d(\ell+1)) + \eta d(\ell+1)/2 \right).$$

Notice that

$$\frac{C_{\ell,\lambda,0} C_{\ell,\lambda,2\mu}}{C_{\ell,\lambda,\mu}^2} = 1,$$

and thus

$$W := \mathbb{E}_P[(Q/P)^2] = \frac{Z_\ell(\lambda,0)Z_\ell(\lambda,2\mu)}{Z_\ell(\lambda,\mu)^2} = \frac{\widetilde{Z}_\ell(\lambda,0)\widetilde{Z}_\ell(\lambda,2\mu)}{\widetilde{Z}_\ell(\lambda,\mu)^2},$$

where

$$\widetilde{Z}_\ell(\lambda,\eta) := \frac{Z_\ell(\lambda,\eta)}{C_{\ell\lambda,\eta}} = \sum_{k=0}^{d} e^{-2\lambda\left( d'k - k^2 \right) - 2\eta(\ell+1)k} \sum_{\substack{j_1,\dots,j_B \\ \sum j_i = k}} \prod \binom{\ell+1}{j_i} \cdot e^{-2(\lambda-\eta)\sum j_i^2}.$$

Let $T_k$ be the $k^{\text{th}}$ term in the above. It holds that $T_k = T_{d-k}$. Indeed, the original terms are invariant under the map $x \mapsto -x$, and for $j = (j_1, \dots, j_B)$, this maps to $(\ell+1)\mathbf{1} - j$ which has the sum $d - k$.

Further, since

$$\sum j_i^2 \leq \max_i(j_i) \sum j_i \leq (\ell+1) \sum j_i,$$

it holds that each term, which depends on $\eta$ as $e^{-2\eta((\ell+1)\sum j_i - \sum j_i^2}$ decreases as $\eta$ increases (or equivalently, $\frac{\partial}{\partial \eta}\widetilde{Z}_\ell(\lambda,\eta) \leq 0$)

Due to the above, for $\mu > 0$,

$$\rho_1 := \frac{\widetilde{Z}_\ell(\lambda, 0) - \widetilde{Z}_\ell(\lambda, \mu)}{\widetilde{Z}_\ell(\lambda, \mu)} \geq 0$$

$$\rho_2 := \frac{\widetilde{Z}_\ell(\lambda, 2\mu) - \widetilde{Z}_\ell(\lambda, \mu)}{\widetilde{Z}_\ell(\lambda, \mu)} \leq 0,$$

yielding,

$$W = \frac{\widetilde{Z}_\ell(\lambda, 0)\widetilde{Z}_\ell(\lambda, 2\mu)}{\widetilde{Z}_\ell(\lambda, \mu)^2} \leq 1 + \rho_1 + \rho_2.$$

(For $\mu < 0$, the signs of both $\rho_1$ and $\rho_2$ are flipped, giving the same bound.)

We now offer control on $\rho_1 + \rho_2$, to complete the argument. To this end, note that

$$1 - 2e^{-2\mu\left((\ell+1)k - \sum j_i^2\right)} + e^{-4\mu\left((\ell+1)k - \sum j_i^2\right)} = \left(1 - e^{-2\mu\left((\ell+1)k - \sum j_i^2\right)}\right)^2,$$

and thus

$$\widetilde{Z}_\ell(\lambda, \mu)(\rho_1 + \rho_2) = \sum_{k=1}^{d-1} \sum_{j:\sum j_i = k} \prod \binom{\ell+1}{j_i} e^{-2\lambda(d'k - k^2 + \sum j_i^2)} \left(1 - e^{-2\mu\left((\ell+1)k - \sum j_i^2\right)}\right)^2$$

$$\leq 2 \sum_{k=1}^{\lfloor d/2 \rfloor} \sum_{j:\sum j_i = k} \prod \binom{\ell+1}{j_i} e^{-2\lambda(d'k - k^2 + \sum j_i^2)} \left(1 - e^{-2\mu\left((\ell+1)k - \sum j_i^2\right)}\right)^2,$$

where we have used the symmetry of the $T_k$s above.

We argue below that the first term in the above strongly dominates all subsequent terms.

**Lemma 35.** *If $\sum j_i = k \in [2 : \lfloor d/2 \rfloor]$, $\ell + 1 \leq d/4$ and $\lambda(d-4) \geq 3\log(d)$, then*

$$\prod \binom{\ell+1}{j_i} e^{-2\lambda(d'k - k^2 + \sum j_i^2)} \leq \frac{1}{d^k} e^{-2\lambda d'}.$$

Using the above, along with $\sum j_i^2 \geq \sum j_i$ and the fact that the number of $B$-tuples of whole numbers that sum up to $k$ is at most $\binom{k+B-1}{k} \leq (eB)^k \leq d^k$, we immediately have

$$\widetilde{Z}_\ell(\lambda, \mu)(\rho_1 + \rho_2) \leq 2d e^{-2\lambda d'} \sum_{k=1}^{d/2} \left(1 - e^{-2\mu\ell k}\right)^2.$$

We bound the sum above in two ways - firstly, each term is $\leq 1$, and so the sum is at most $d/2$. Further, using $1 - e^{-x} \leq x$, the sum is at most $4\sum \mu^2 \ell^2 k^2 \leq \mu^2 d^5$. This gives ,

$$\widetilde{Z}_\ell(\lambda, \mu)(\rho_1 + \rho_2) \leq 2d^2 \min(1, \mu^2 d^4) e^{-2\lambda(d-1-\ell)}$$

The bound on $W$ now follows since $\widetilde{Z}_\ell(\lambda, \mu) \geq 2$ trivially. $\qquad\square$

*Proof of Lemma 35.* This is essentially the same as Lemma 33, and may be proved similarly. $\qquad\square$

### F.2.5 The Clique versus the Empty Graph in High Temperatures

*Proof of Proposition 22.* This proof heavily relies on techniques we encountered in [CNL18]. The principal idea is via the following representation of the law of an Ising model with uniform edge weights, and the subsequent expression (and upper bound) for its partition function, both of which we encountered in the cited paper.

Let $\tau = \tanh(\mu)$. Then the law of the Ising model on a $m$-vertex graph $G$ with uniform weights $\alpha$ is

$$P(X = x) = \frac{\prod_{(i,j)\in G}(1 + \tau X_i X_j)}{2^m \mathbb{E}_0[\prod_{(i,j)\in G}(1 + \tau X_i X_j)]},$$

where $\mathbb{E}_0$ denotes expectation with respect to the uniform law on $\{-1,1\}^m$. This is shown by noticing that $\exp(x) = \cosh(x)(1 + \tanh(x))$, and then observing that for $x = \mu X_i X_j$, since $X_i X_j = \pm 1$, the same is equal to $\cosh(\mu)(1 + \tanh(\mu) X_i X_j)$. The $\cosh(\mu)$ term is fixed for all entries, and thus vanishes under the normalisation. The denominator is simply a restatement of $\sum_{\{-1,1\}^m} \prod_{(i,j) \in G}(1 + \tau X_i X_j)$.

Let the denominator of the above be denoted $2^m \Phi(\tau; G)$. We further have the expansion

$$\Phi(\tau; G) = \sum_{u \geq 0} \mathscr{E}(u, G) \tau^u,$$

where $\mathscr{E}(j, G)$ denotes the number of 'Eulerian subgraphs of $G$', where we call a graph Eulerian if each of its connected components is Eulerian (and recall that a connected graph is Eulerian if and only if each of its nodes has even degree). This arises by expanding the above product out to get

$$\Phi(\tau; G) = \sum_{u \geq 0} \tau^u \cdot \sum_{\text{choices of } u \text{ edges } (i_1, j_1), (i_2, j_2), \dots (i_u, j_u)} \mathbb{E}_0[X_{i_1} X_{j_1} \dots X_{i_u} X_{j_u}].$$

Now, due to the independence, if any node of the $X_i$s or the $X_j$s appears an odd number of times in the product, the expectation of that term under $\mathbb{E}_0$ is zero. If they all appear an even number of times, the value is of course 1. Thus the inner sum, after expectation, amounts to the number of groups of $u$ edges such that each node occurs an even number of times in this set of edges, which corresponds to the number of Eulerian subgraphs of $G$, defined in the above way.

A further subsidiary lemma controls the size of $\mathscr{E}(u, G)$ as follows, where we abuse notation and use $G$ to denote the adjacency matrix of the graph $G$.

$$\mathscr{E}(u, G) \leq (2\|G\|_F)^u.$$

The idea behind this is to first control the number of length-$v$ closed walks in a graph, by noticing that the total number of length $v$ walks from $i$ to $i$ is $(G^v)_{i,i}$, summing which up gives an upper bound on the number of closed length $v$ walks of $\text{Tr}(G^v) \leq \|G\|_F^v$. Next, we note that to get an Eulerian subgraph of $G$ with $u$ edges, we can either take a closed walk of length $u$ in $G$, or we can add a closed walk of length $v \leq u - 2$ to an Eulerian subgraph with $u - v$ edges. This yields a Grönwall-style inequality that the authors solve inductively. Please see [CNL18, Lemma A.1].

Now, let $P$ be the Ising model $K_m$ with uniform weight $\alpha$, and let $Q$ be the Ising model on the empty graph on $m$ nodes. Using the above expression for the law of an Ising model, we have

$$1 + \chi^2(Q\|P) = \mathbb{E}_Q[Q/P] = \mathbb{E}_0[\prod_{i<j}(1 + \tau X_i X_j)]\mathbb{E}_0[\prod_{i<j}(1 + \tau X_i X_j)^{-1}],$$

which, by multiplying and dividing each term in the second expression by $1 - \tau X_i X_j$, and noting that $X_i^2 X_j^2 = 1$, may further be written as

$$1 + \chi^2(Q\|P) = \mathbb{E}[\prod_{i<j}(1 + \tau X_i X_j)]\mathbb{E}\left[\frac{\prod_{i<j}(1 - \tau X_i X_j)}{(1 - \tau^2)^{-\binom{m}{2}}}\right]$$
$$= \Phi(\tau; K_m)\Phi(-\tau; K_m)(1 - \tau^2)^{-\binom{m}{2}}.$$

Since the above expression is invariant under a sign flip of $\tau$, we may assume, without loss of generality, that $\tau \geq 0$. Next, notice, due to the expansion in terms of $\mathscr{E}$ of $\Phi$, that $\Phi(-\tau; K_m) \leq \Phi(\tau; K_m)$ for $\tau \geq 0$. Further, for $\tau \geq 0$, using the bound on $\mathscr{E}(u, G)$,

$$\Phi(\tau; K_m) \leq \mathscr{E}(0; K_m) + t\mathscr{E}(1; K_m) + t^2\mathscr{E}(2; K_m) + \sum_{u \geq 3}(2t\|K_m\|_F)^u.$$

Now notice that $\mathscr{E}(0; K_m) = 1$, and $\mathscr{E}(1; K_m) = \mathscr{E}(2; K_m) = 0$. The first of these is because there is only a single empty graph, while the other two follow since $K_m$ is a simple graph. Further, $\|K_m\|_F = \sqrt{m(m-1)} \leq m$. Thus, we have

$$\Phi(\tau; K_m) \leq 1 + \sum_{u \geq 3}(2tm)^u.$$

Now, since $2\tanh(\alpha)m \le 2\alpha m \le 1/16 < 1/2$, we sum up and bound the geometric series to conclude that $\Phi(\tau; K_m) \le 1 + 16(tm)^3 \le 1 + (tm)^2$, and as a consequence,

$$\Phi(\tau; K_m)^2 \le (1 + (tm)^2)^2 \le 1 + 3(tm)^2 \le \exp\left(3(tm)^2\right).$$

Further, since $\tau m < 1/32$, and $m \ge 1$, we have $\tau < 1/32$, which in turn implies that $(1 - \tau^2)^{-1} \le \exp\left(2\tau^2\right)$. Thus, we find that

$$1 + \chi^2(P\|Q) \le \exp\left(3(\tau m)^2\right) \cdot (\exp\left(2\tau^2\right))^{m^2/2} \le \exp\left(4(\tau m)^2\right) \le 1 + 8(\tau m)^2,$$

where the final inequality uses the fact that for $x < \ln(2)$, $e^x \le 1 + 2x$, which applies since $4(\tau m)^2 \le 4/(32)^2 < \ln(2)$. $\qquad\square$

It is worth noting that Proposition 21 is also shown in the above framework by [CNL18]. The main difference, however, is that in the $\chi^2$ computations, the square of $\prod(1 + \tau X_i X_j)$ appears. The technique the authors use is to extend the notion of $\mathscr{E}$ to multigraphs, and show the same expansion for these, along with the same upper bound for $\mathscr{E}(u, G)$, this time with the entries of $G$ denoting the number of edges between the corresponding nodes.