[Reviews · NeurIPS 2020]

Review 1

Summary and Contributions: This paper presents minimax bounds on sample complexity to learn changes in a network are the same as learning the network directly when the change is small enough.

Strengths: Soundness: The minimax proof follows the traditional LeCam approach. The proof is based on a lifting idea that appears novel. Significance and novelty: this work is novel in that it seems to contradict other results in the literature stating that learning changes is easier than learning the network directly. Relevance to NeurIPS: network learning is relevant to NeurIPS community.

Weaknesses: Soundness: no weakness that I can see Significance and novelty: it would be good if the authors could discuss potential generalizations to networks beyond the Ising model. Relevance to NeurIPS: no weakness

Correctness: The proofs appear correct.

Clarity: This paper is clearly written.

Relation to Prior Work: Prior work is discussed.

Reproducibility: Yes

Additional Feedback:


Review 2

Summary and Contributions: This paper considers the problems of testing and learning differences in the edge structures between two degree bounded Ising models from an information theoretic perspective. The lower bounds on the sample complexities of three problems are established: 1. a testing problem to test if two Ising models are s-separated (differ in at least s edges) 2. a learning problem to approximately learn the difference between the edge structures of two s-separated Ising models 3. structure learning of an Ising model with approximate recovery criterion. The bounds for the aforementioned problems are compared against an achievability bound on the sample complexity of approximate structure learning using an ML decoder. The paper also provides results for a specific case of forest structured Ising models.

Strengths: The problems studied in this paper are well motivated as the statistical limits on the sample complexity of functional inference over two or more graph models have not been established for a wide class of problems. This paper tries to make a contribution in that direction by considering the detection and learning of changes between Ising models. The analysis primarily uses standard information theoretic workhorses like Le Cam’s method and Chi-squared based bounds with novel and non-trivial ensemble constructions to derive the results. The paper establishes theoretically that the lower bounds on the detection and learning of changes in the Ising models have approximately the same scaling behavior as that of structure learning for a wide range of regimes. This is in contrast to the claims in several existing algorithmic approaches that imply that recovery of sparse changes is possible with smaller sample complexity than structure learning of the complete graphs. For forest structured graphs, it is theoretically established that detecting large changes has lower sample complexity than that for learning the changes and structure learning in Ising models, which is of practical significance.

Weaknesses: Overall I like the paper technically, but there is scope for elaboration on some aspects. 1. Proposition 1 is not intuitive to me from the exposition provided in the paper. Essentially, SL, EoF and GoF are separate problems with different inputs (for instance , SL takes the samples from only one graph as an input in contrast to EoF and GoF) and the argument that solving one of those problems leads to the solution to another problem is not very convincing for establishing the order of difficulties of the problems. 2. The paper lacks any experimental validation of its results. Specifically, experiments on forest structured graphs to illustrate the deviation between the sample complexities of testing problem and the learning problems would have been informative.

Correctness: The theoretical claims are correct to the best of my knowledge.

Clarity: The paper is very well written with great emphasis on the interpretation of different results. The sample complexity results are elaborated upon in different regimes. The novelties in the proof techniques are also emphasized upon in the paper.

Relation to Prior Work: There are not many information-theoretic works in this domain. The authors discuss the existing works on detection of sparse changes in Ising models and compare their results against them.

Reproducibility: Yes

Additional Feedback:


Review 3

Summary and Contributions: This paper considers goodness-of-fit testing and error-of-fit testing for Ising graphical models. The test is meant to discover whether there are s edges that are different. It is shown, somewhat surprisingly, that the sample complexity result for these two tests, in some regimes of s, is just as high as that for learning the entire structure itself.

Strengths: The main result is encapsulated in Theorem 3 where lower bounds on n_{GoF} and n_{EoF} are obtained. It is shown that in some regimes of s, these lower bounds match the ML one of Santhanam and Wainwright. I did not go through all of the long supplementary material. However, what I did go through looks interesting and novel. In particular, the use of information-theoretic techniques (Le Cam's method), the construction of different instances (e.g., based on Emmenthaler cliques) to compute the lower bounds is interesting and non-trivial.

Weaknesses: I do not see any major limitations of this work. However, one aspect that could improve this work is to add some numerical experiments to check whether the dependencies on the various parameters is sharp. This can be done for, for example, Theorem 7. This would add to the value and impact of the paper.

Correctness: I didn't have time to go through all the material in the supplementary material. However, what I did go through looks correct. I am not an expert in this area.

Clarity: Yes.

Relation to Prior Work: This is fine.

Reproducibility: No

Additional Feedback: NIL

[Author Response · NeurIPS 2020]

Our thanks to each of the reviewers for their work and their suggestions. We discuss their questions below, and we will include such discussions in a final version of the paper.

**Generalisations beyond the Ising Model** (Reviewer 1) While the generic techniques of Le Cam's method, and in our case the lifting trick of §3.1 should naturally extend to any graphical model, most of the calculations that offer specific control on $\chi^2$-divergences are strongly affected by the law of the graphical model under consideration. It is, of course, for this reason that most papers in this space commit to studying either Ising models or Gaussian MRFs (GMRFs), in that these are both natural models with relatively tractable calculations. For the particular choice of GMRFs, we feel that the same constructions should extend naturally to show similar results w.r.t. the separation of the sample complexity of recovery and testing. This is due to the (non-rigorous) intuition that the pairwise properties of GMRFs behave similarly to high-temp Ising models (with obvious caveats), where we have established these effects.

**Experimental Validation in the Tree Setting** (Reviewers 2 and 3) This is a very valid point, and should be tractable. We will include such an experiment in a final version of the paper.

**Elaboration of Proposition 1** (Reviewer 2) The main thing that allows comparison is that risks for $\mathbb{SL}$ and $\mathbb{EOF}$ are defined in terms of the probability of an error event. The intended reductions are as follows.

1. Suppose we have a $\lfloor s/2 \rfloor$-approximate structure learner with risk $\delta$. Then we can construct the following $\mathbb{EOF}$ estimator with the same sample costs: Take a sample from $Q$, and pass it to the structure learner. With probability at least $1 - \delta$, this gives a graph $\hat{G}$ that is at most $\lfloor s/2 \rfloor$-separated from $G(Q)$. Now compute $G(P) \triangle \hat{G}$ ($G(P)$ is determined because $P$ is given to the $\mathbb{EOF}$ tester). By the triangle inequality applied to the adjacency matrices of the graphs under the Hamming metric, this identifies $G(P) \triangle G(Q)$ up to an error of $\lfloor s/2 \rfloor$.

2. For $\mathbb{GOF}$, we can use a scheme for $\mathbb{EOF}$ as follows: take enough samples so that $\mathbb{EOF}$ can be solved with risk at most $\delta/2$. Take the samples from $Q$ and pass them to the $\mathbb{EOF}$ solver. With probability at least $1 - \delta/2$, if $G(P) = G(Q)$, then this solver must output a graph with at most $s/2$ edges, which if they are separated, this must output a graph with at least $s/2$ edges. So, thresholding the number of edges in this output yields a $\mathbb{GOF}$ tester. Net risk is $\delta/2$ for size and for power, giving sum risk $\delta$.

We must thank you for bringing this up, because writing out this proof shows that there are small bookkeeping errors in the definitions (the complexities of $\mathbb{SL}$ and $\mathbb{EOF}$ should be defined as the level needed to get error $1/8$ and not $1/4$, since going from $\mathbb{EOF}$ to $\mathbb{GOF}$ is doubling the risk in the above, and the $\mathbb{EOF}$ risk should penalise errors of at least $s/2 - 1$ instead of $s/2$ to make the thresholding work out correctly). These do not affect the results besides small constant corrections, but are important to get right.

[Meta-Review · NeurIPS 2020]

Congratulations on a nice piece of work. The reviewers have made some expository comments and have requested experimental validation. It would be great if those made it into the final version.